# Certain Head, Uncertain Tail: Expert-Sample for Test-Time Scaling in Fine-Grained MoE

**Yuanteng Chen** [1 2 3]  **Peisong Wang**[⊠ 1 3]  **Nanxin Zeng** [3]  **Yuantian Shao** [1 4]
**Shuang Qiu** [5]  **Gang Li** [1]  **Jing Liu** [1 2 3]  **Jian Cheng**[⊠ 1 2 3]

## Abstract

Test-time scaling improves LLM performance by generating multiple candidate solutions, yet token-level sampling requires temperature tuning that trades off diversity against stability. Fine-grained MoE, featuring hundreds of well-trained experts per layer and multi-expert activation per token, offers an unexplored alternative through its rich routing space. We empirically characterize fine-grained MoE routing and uncover an informative pattern: router scores exhibit a certain head of high-confidence experts followed by an uncertain tail of low-confidence candidates. While single-run greedy accuracy remains stable when fewer experts are activated, multi-sample pass@n degrades significantly—suggesting that the certain head governs core reasoning capability while the uncertain tail correlates with reasoning diversity. Motivated by these findings, we propose Expert-Sample, a training-free method that preserves high-confidence selections while injecting controlled stochasticity into the uncertain tail, enabling diverse generation without destabilizing outputs. Evaluated on multiple fine-grained MoE models across math, knowledge reasoning, and code tasks, Expert-Sample consistently improves pass@n and verification-based accuracy. On Qwen3-30B-A3B-Instruct evaluated on GPQA-Diamond with 32 parallel samples, pass@32 rises from 85.4% to 91.9%, and accuracy improves from 59.1% to 62.6% with Best-of-N verification.

---

[1]Institute of Automation, Chinese Academy of Sciences [2]Zhongguancun Academy, Beijing, China [3]School of Artificial Intelligence, University of Chinese Academy of Sciences [4]Nanjing University of Science and Technology [5]City University of Hong Kong. Correspondence to: Peisong Wang <peisong.wang@nlpr.ia.ac.cn>, Jian Cheng <jcheng@nlpr.ia.ac.cn>.

*Proceedings of the 43rd International Conference on Machine Learning*, Seoul, South Korea. PMLR 306, 2026. Copyright 2026 by the author(s).

## 1. Introduction

Mixture-of-Experts (MoE) (Artetxe et al., 2022) has become one of the most effective approaches for scaling model parameters through sparse expert activation. Recently, fine-grained MoE designs (Dai et al., 2024) featuring hundreds of well-trained experts per layer and multi-expert activation per token have gained prominence. Models such as DeepSeek-R1 (DeepSeek-AI, 2025), GPT-OSS (OpenAI, 2025), and the Qwen3-MoE (Yang et al., 2025) series adopt this architecture and significantly outperform earlier MoE models like Mixtral-8x7B (Jiang et al., 2024) that use fewer experts. Yet existing works (Jin et al., 2024; Yan et al., 2025) have focused almost exclusively on training efficiency and serving optimization, leaving the inference-time potential of the rich routing space largely unexplored.

Test-time scaling (Brown et al., 2024) has emerged as a powerful paradigm for improving LLM performance, particularly on complex reasoning tasks. By generating multiple candidate solutions and selecting the best one through verification (Irvine et al., 2023) or majority voting (Wang et al., 2023), models can achieve accuracy far beyond single-run inference. However, current approaches predominantly rely on token-level sampling to produce diverse candidates, where temperature serves as the primary control knob. This creates a well-known dilemma: higher temperatures increase diversity but degrade individual sample quality, while lower temperatures preserve quality but limit exploration of the solution space (Pipis et al., 2025). This motivates the search for alternative sources of diversity that can maintain sample quality while enabling effective exploration.

In this paper, we investigate whether the routing mechanism in fine-grained MoE can serve as an alternative source of diversity for test-time scaling. We begin with an empirical study and observe (in section 2) that when the number of activated experts is significantly reduced, single-run greedy decoding accuracy remains surprisingly stable, yet multi-sample pass@n performance degrades substantially. This asymmetry prompts us to take a closer look at router score distributions, where we uncover an informative pattern: a certain head consisting of a small number of high-confidence experts, followed by an uncertain tail of many

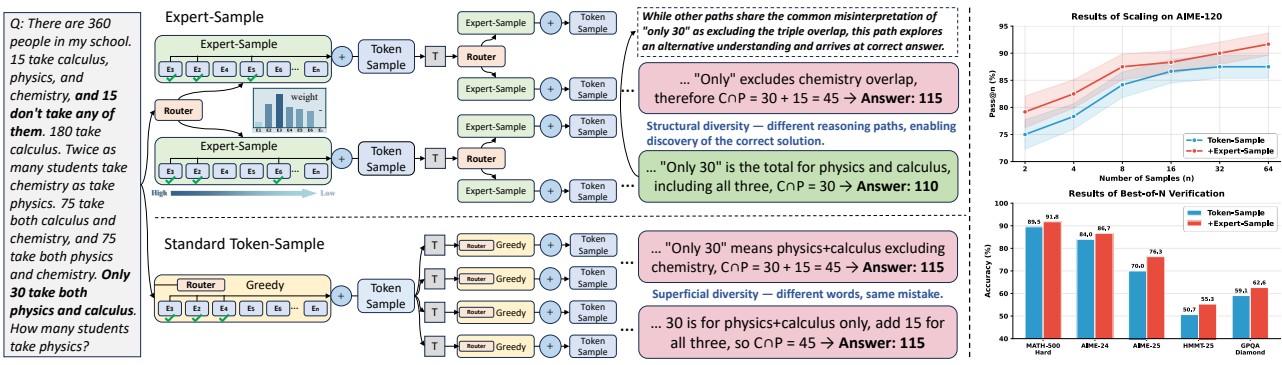

*Figure 1.* Overview of Expert-Sample. **Left**: Illustration of the Expert-Sample mechanism with an example from MATH-500, showing how Expert-Sample achieves structural diversity to discover the correct answer, while standard token-sample produces only superficial diversity. **Right**: Pass@n scaling improvements (upper) and accuracy gains with Best-of-N verification (lower) on Qwen3-30B-A3B-Instruct.

experts with relatively uniform weights. The certain head appears sufficient for deterministic generation, while uncertain tail enables diverse solution paths under parallel sampling.

These findings reveal an opportunity: we can maintain generation stability by preserving the certain head while injecting diversity through stochastic sampling in the uncertain tail. This routing-level approach provides an additional dimension for diversity that complements rather than replaces token-level sampling.

Building on this insight, we propose Expert-Sample, a simple yet effective method for test-time scaling in fine-grained MoE models. At each layer, Expert-Sample deterministically retains the top-ranked experts with high-confidence routing weights (*e.g.*, $E_3, E_2$ in Figure 1), then samples the remaining activated experts from a specified rank range (*e.g.*, $E_4, E_5, E_6, \ldots$) using temperature-scaled router logits, while preserving the original gating weights for expert output aggregation.

As illustrated in the case study (Figure 1, Left), this mechanism enables structurally diverse reasoning paths that discover the correct answer, while standard token-sample produces only superficial diversity with the same underlying mistake. Crucially, Expert-Sample acts as a plug-and-play sampling strategy requiring no architectural modification or additional training, and complements rather than conflicts with token-level sampling. As shown in Figure 1 (Right), on Qwen3-30B-A3B-Instruct, Expert-Sample on top of standard token-sample significantly improves both pass@n accuracy (upper) and verification-based accuracy (lower) across multiple benchmarks.

We evaluate Expert-Sample on multiple fine-grained MoE models including Qwen3-MoE, GPT-OSS and Ling-Lite-1.5 (Team, 2025) across diverse tasks spanning math reasoning, knowledge-intensive reasoning, and code generation. Compared to token-level sampling, our experiments demonstrate that Expert-Sample consistently improves pass@n accuracy under multi-sample generation, indicating stronger

scaling potential. Furthermore, Expert-Sample composes favorably with existing selection strategies such as Best-of-N and majority voting, boosting verification-based accuracy and translating to practical performance gains.

## 2. Motivation

### 2.1. Expert Reduction Does Not Hurt Greedy Accuracy but Degrades Multi-Sample Pass@n Performance

To understand how expert selection affects generation behavior in fine-grained MoE, we conduct a preliminary experiment where we reduce the number of activated experts at each layer during inference. We evaluate five representative fine-grained MoE models covering most of the latest popular architectures: Qwen3-Next-80B-A3B-Instruct, Qwen3-30B-A3B-Instruct, GPT-OSS-20B, Ling-Lite-1.5, and DeepSeek-V2-Lite-Chat (DeepSeek-AI et al., 2024). Experiments are conducted on GPQA-Diamond (Rein et al., 2023), a professional knowledge reasoning benchmark, and AIME-120, which contains 120 competition-level math problems from 2022 to 2025. Crucially, we use greedy decoding to eliminate the interference of sampling randomness, allowing us to more directly measure the impact of expert selection on the model's core reasoning capability.

As shown in Figure 2(a), greedy decoding accuracy remains remarkably stable even when the number of activated experts is reduced to half of the default top-$k$. This holds consistently across all five models and both benchmarks, suggesting that the top half of experts ranked by routing weights are sufficient for deterministic generation and preserving the model's core reasoning capability.

Given that reducing experts does not hurt core reasoning ability under greedy decoding, a natural question arises: *since the number of activated experts is fixed during training, what role do the remaining selected experts play?* To investigate, we further explore how expert reduction affects pass@n accuracy when token-level sampling is introduced.

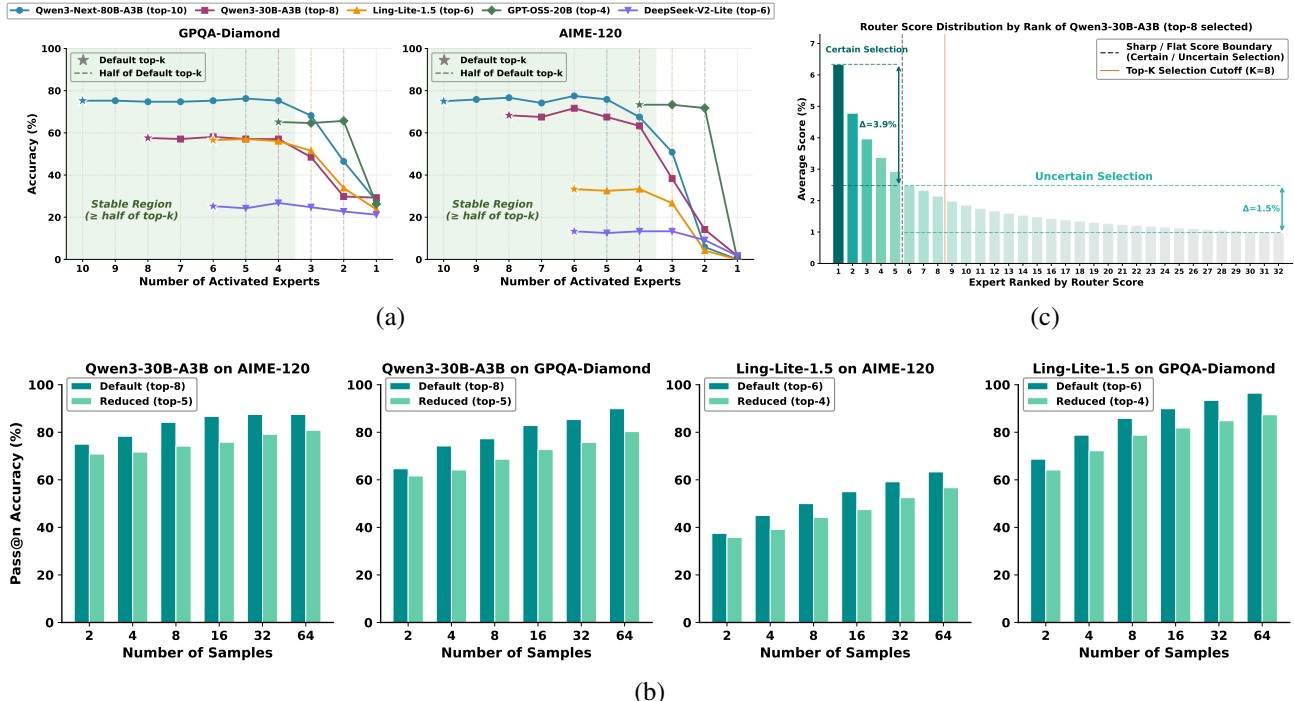

*Figure 2.* Empirical findings and motivation for Expert-Sample. (a) Greedy decoding accuracy remains stable when reducing activated experts to half of the default top-k. (b) Pass@n accuracy degrades substantially when expert count is reduced, suggesting the uncertain tail is critical for diverse exploration. (c) Router score distribution for the top 32 ranked positions reveals a certain head with high-confidence experts and an uncertain tail with uniform scores (full distribution is in Appendix E). All Qwen3 models shown are Instruct versions.

Specifically, we take Qwen3-30B-A3B-Instruct and Ling-Lite-1.5 as examples, generate 64 parallel samples per problem using standard temperature sampling ($T = 0.7$), and compute pass@n at various sample sizes. The number of activated experts is set to approximately half of the default ($k/2+1$), a configuration that causes negligible degradation in greedy accuracy as shown in Figure 2(a).

As illustrated in Figure 2(b), pass@n accuracy drops substantially under the reduced expert configuration. This asymmetry reveals that the additional experts beyond the certain top few contribute minimally to deterministic generation but play a critical role in enabling diverse outputs under sampling—in other words, they expand the space of possible reasoning paths and contribute to the diversity.

### 2.2. A Closer Look at Router Score Distribution

The findings above prompt us to examine router score distributions more closely. If reducing experts hurts diversity but not greedy accuracy, there must be a structural difference between top-ranked versus lower-ranked experts.

To investigate this, we take Qwen3-30B-A3B-Instruct as an example, which has 128 experts per layer with 8 experts activated per token by default. We collect the router scores (i.e., the router outputs after softmax normalization) for all tokens across both prefill and decode stages on GPQA-Diamond

and AIME-120. For each layer, we rank the experts by their router scores, and then average the scores at each rank position across all tokens and all layers. Figure 2(c) visualizes the resulting distribution for the top 32 ranked positions.

The distribution reveals a clear pattern: a sharp certain head followed by a flat uncertain tail. Within the top-ranked positions, router scores vary dramatically—the gap from rank 1 to rank 5 alone exceeds 3.9%. In contrast, from approximately the 5th position onward, the scores become remarkably uniform—the cumulative difference from rank 5 to rank 32 is less than 1.5%, even though rank 32 extends to four times the default top-$k$ selection. Notably, there exists a distinct boundary between the certain head and the uncertain tail, which roughly coincides with half of the default top-$k$ selection. This structural property explains the asymmetry observed in Section 2.1: the certain head captures the essential computation for deterministic reasoning, while the flat distribution in the uncertain tail indicates that these experts are not strongly preferred over one another, making them natural candidates for introducing diversity through stochastic selection. In Appendix E, we present more comprehensive weight distribution results.

### 2.3. Decoupling Stability and Diversity through Routing

Together, these observations reveal a key opportunity for test-time scaling. The certain head and uncertain tail in fine-

grained MoE routing naturally decouple the requirements for stability and diversity—two goals that are notoriously difficult to balance in token-level sampling.

By deterministically preserving the high-confidence experts in the certain head, we can maintain stable generation quality comparable to greedy decoding. By introducing controlled stochasticity in the uncertain tail, we can explore diverse reasoning paths without destabilizing individual outputs.

## 3. Method

In this section, we present **Expert-Sample**, a training-free, plug-and-play inference-time sampling strategy for fine-grained MoE models. We first describe the standard expert selection mechanism and then introduce our method that injects stochasticity into the uncertain tail of the expert distribution. Subsequently, we validate Expert-Sample through comprehensive experiments evaluating both stability and diversity, and finally discuss hyperparameter choices.

### 3.1. Standard Expert Selection

In fine-grained MoE architectures, each token's hidden state $\mathbf{h} \in \mathbb{R}^d$ is passed through a gating network that produces logits $\mathbf{g} = \mathbf{h} \cdot \mathbf{W}_g \in \mathbb{R}^n$, where $\mathbf{W}_g \in \mathbb{R}^{d \times n}$ is the gating weight matrix and $n$ is the total number of experts. These logits are normalized via softmax to obtain routing probabilities $\mathbf{p} = \mathrm{softmax}(\mathbf{g})$. The top-$k$ experts with the highest probabilities are selected and their weights renormalized:

$$\mathcal{S} = \text{top-}k(\mathbf{p}), \quad \tilde{p}_i = \frac{p_i}{\sum_{j \in \mathcal{S}} p_j}, \quad \forall i \in \mathcal{S} \quad (1)$$

The final MoE output is the weighted sum of the selected experts' outputs:

$$\mathbf{o} = \sum_{i \in \mathcal{S}} \tilde{p}_i \cdot \mathrm{Expert}_i(\mathbf{h}) \quad (2)$$

This standard selection is essentially a **greedy choice**—analogous to greedy decoding in token generation—where the same $k$ experts are deterministically activated given the same input. As our analysis in Section 2 reveals, this greedy selection preserves the certain head that is essential for core reasoning, but it also rigidly fixes the uncertain tail, eliminating a natural source of diversity.

### 3.2. Expert-Sample

Based on our observation that the certain head should remain stable while the uncertain tail can tolerate stochasticity, we propose Expert-Sample with three hyperparameters: $k_{\mathrm{keep}}$ (number of top experts to keep deterministically), temperature $\tau$, and sampling range $r$.

The procedure works as follows:

1. **Preserve the certain head:** The top $k_{\mathrm{keep}}$ experts ranked by their gating weights are always selected, ensuring that the core reasoning path remains intact.

2. **Sample from the uncertain tail:** For the remaining $k - k_{\mathrm{keep}}$ slots, we sample from a candidate pool of experts ranked from position $k_{\mathrm{keep}} + 1$ to $r$ (where $r > k$). Let $\mathbf{g}_{[k_{\mathrm{keep}}+1:r]}$ denote the gating logits of these candidates. We apply temperature scaling and softmax to obtain scores, and use Gumbel-softmax sampling:

$$\mathbf{p}' = \mathrm{softmax}\left(\frac{\mathbf{g}_{[k_{\mathrm{keep}}+1:r]}}{\tau}\right) \quad (3)$$

$$\mathcal{S}_{\mathrm{tail}} = \text{Gumbel-top-}(k - k_{\mathrm{keep}})(\mathbf{p}') \quad (4)$$

The Gumbel-softmax trick ensures that the probability of selecting each candidate is proportional to its score while requiring only a single forward pass, thus preserving inference efficiency.

3. **Renormalize with original weights:** The final selected set is $\mathcal{S} = \mathcal{S}_{\mathrm{head}} \cup \mathcal{S}_{\mathrm{tail}}$. Crucially, we retrieve original gating weights for all selected experts and renormalize.

This design ensures that (1) the certain head is never perturbed, maintaining stability on problems the model can already solve; (2) the uncertain tail is stochastically sampled, enabling diverse reasoning paths; and (3) the original weight magnitudes are preserved for the final weighted sum, respecting the model's learned preferences. Notably, since $k_{\mathrm{keep}}$ and other hyperparameters remain constant across the batch, all operations are fully vectorized, incurring negligible overhead on overall inference speed.

Figure 1 illustrates how Expert-Sample works. Standard token sampling introduces diversity solely at the output distribution level after the forward pass. Expert-Sample augments this by injecting additional diversity earlier in the computation—at the expert routing level within each MoE layer. Since this diversity originates from the expert level, Expert-Sample does not conflict with token-level sampling and can achieve sufficient reasoning diversity with normal-temperature sampling, avoiding the instability associated with high-temperature decoding.

### 3.3. Validation: Balancing Stability and Diversity

We validate that Expert-Sample achieves the best of both worlds: maintaining stability on problems the model can reliably solve, while improving diversity and accuracy on challenging problems.

**Experimental Setup.** We use AIME-120 as our testbed and evaluate on Qwen3-30B-A3B-Instruct and Ling-Lite-1.5. For each model, we first perform 5 independent runs per problem under standard token sampling to identify problems the model can reliably solve. For the remaining uncertain

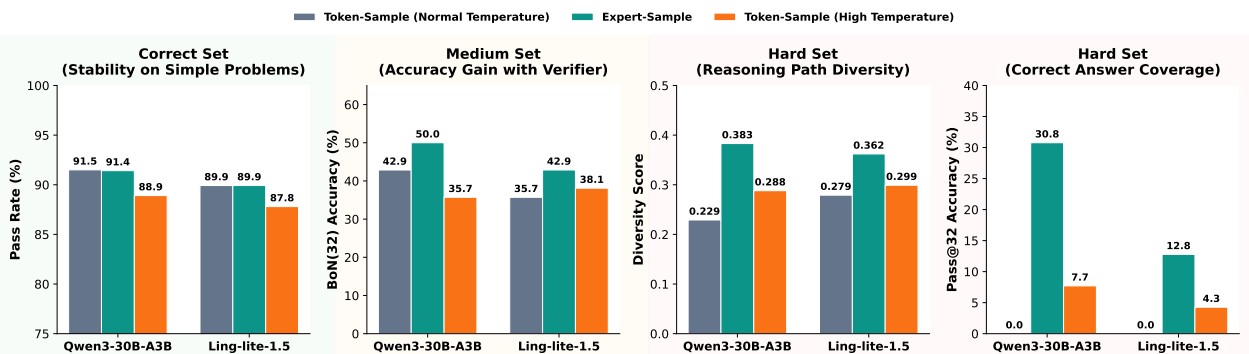

*Figure 3.* Validation results on AIME-120 across three difficulty tiers. For Qwen3-30B-A3B (Instruct), the Correct/Medium/Hard sets contain 79/28/13 problems respectively; for Ling-Lite-1.5, 31/42/47 problems respectively.

problems, we conduct 32 runs to further categorize them based on whether any correct answer is produced. This results in three difficulty tiers, each paired with an evaluation metric tailored to its specific challenge:

1. **Correct Set (Stability):** Problems where the model answers correctly in $\geq 4$ out of 5 runs. These represent problems the model can reliably solve. We measure *pass rate* across 32 runs—higher values indicate that the sampling method maintains the model's ability without introducing harmful variance.

2. **Medium Set (Accuracy with Verification):** From the remaining uncertain problems, those where at least 1 out of 32 runs yields a correct answer. These are problems within the model's capability but requiring multiple attempts. We apply Best-of-N selection using Qwen2.5-Math-PRM (Qwen et al., 2025) as a verifier to score all 32 responses and select the highest-scored answer, evaluating whether the sampling method produces responses more likely to be verified as correct.

3. **Hard Set (Diversity and Coverage):** From the remaining uncertain problems, those with 0 correct answers across 32 runs under standard sampling. For these challenging problems, we evaluate from two perspectives:

   - *Process diversity*: We use DeepSeek-R1 as a judge to evaluate reasoning path similarity between all pairs of responses for each problem. For each pair, the judge first analyzes and identifies the main reasoning steps in both responses, then compares these steps while ignoring differences in final answers, and assigns an integer score from 0 to 5 indicating similarity (higher means more similar). We compute the average similarity across all response pairs, normalize it to $[0, 1]$, and define diversity score as $1 - \text{similarity}$. Detailed prompts and example outputs are provided in Appendix D.
   - *Outcome coverage*: We measure pass@32 accuracy—whether any of the 32 samples produces a cor-

rect answer, to assess if diverse sampling helps discover correct solutions.

**Baselines.** We compare Expert-Sample against token sampling at two temperature settings. For normal-temperature token sampling, we set $T = 0.7$, top-$p = 0.8$, and top-$k = 20$; for high-temperature token sampling, we set $T = 1.3$, top-$p = 0.98$, with no top-$k$ restriction. For Expert-Sample, we use default configuration ($k_{\text{keep}} = \lfloor k/2 \rfloor + 1, \tau = 1.0, r = 4k$) combined with normal-temperature token sampling.

**Results.** Figure 3 presents the results across 3 problem tiers.

*Stability on Correct Set.* Compared to normal-temperature token sampling, high-temperature sampling causes pass rate to drop by 2.6% and 2.1% on the two models respectively, demonstrating the well-known trade-off between diversity and stability. In contrast, Expert-Sample causes only a 0.1% drop on Qwen3-30B-A3B-Instruct and no drop on Ling-Lite-1.5, confirming that expert-sample does not destabilize the model on problems it can already solve.

*Accuracy Gains on Medium Set.* High-temperature sampling shows inconsistent behavior across models—improving accuracy on Qwen3-30B-A3B-Instruct but degrading it on Ling-Lite-1.5. In contrast, Expert-Sample provides consistent gains and achieves the highest BoN(32) accuracy on both models when combined with Best-of-N verification, improving over normal-temperature sampling by 7.1% on Qwen3-30B-A3B-Instruct and 7.2% on Ling-Lite-1.5.

*Diversity and Coverage on Hard Set.* When examining reasoning path diversity, while high-temperature sampling shows some improvement over normal-temperature, Expert-Sample brings substantially larger gains—achieving diversity scores of 0.383 and 0.362 compared to 0.288 and 0.299 for high-temperature sampling. This translates to concrete accuracy improvements: on pass@32, Expert-Sample enables Qwen3-30B-A3B-Instruct to solve 30.8% of previously unsolvable problems (vs. 7.7% for high-temperature), and Ling-Lite-1.5 to solve 12.8%, which means 6 out of 47

previously impossible problems are now solvable—a direct consequence of enhanced reasoning diversity.

These results demonstrate that Expert-Sample successfully decouples stability from diversity: it preserves the model's reliability on tractable problems while substantially expanding its problem-solving coverage on challenging ones.

### 3.4. Hyperparameter Analysis

Expert-Sample introduces three hyperparameters: $k_{\text{keep}}$ for stability preservation, temperature $\tau$ and sampling range $r$ for controlling the aggressiveness of stochastic selection.

An important question is whether these hyperparameters require careful tuning for different models. To answer this, we conduct extensive ablation studies on Qwen3-30B-A3B-Instruct and Ling-Lite-1.5 **(detailed in Appendix C)**. The results suggest that Expert-Sample is robust to hyperparameter choices: performance remains stable across a wide range of settings, with the optimal values matching our theoretical expectation from Section 2. We thus recommend the following default configuration that works reliably across all tested models and benchmarks: $k_{\text{keep}} = \lfloor k/2 \rfloor + 1$, $\tau = 1.0$, and $r = 4k$. This setting can be directly applied to any fine-grained MoE model as a drop-in enhancement.

## 4. Experiments

In this section, we first introduce our experimental setup, including models, benchmarks, and implementation details. We then validate that Expert-Sample enables more efficient scaling, as evidenced by consistent pass@n accuracy improvements across different computational budgets. Finally, we combine Expert-Sample with several verification methods to select final answers from multiple responses, demonstrating its practical utility in boosting model accuracy.

### 4.1. Experimental Setup

**Models.** To comprehensively validate the effectiveness of Expert-Sample, we select four fine-grained MoE models spanning a wide range of configurations: total parameters from 16B to 80B, varying numbers of experts, and different activation counts per token. For GPT-OSS-20B, we use the low think-budget configuration. Table 1 summarizes the detailed specifications. This diverse selection ensures that our findings generalize across different architectures.

*Table 1.* Overview of fine-grained MoE models used in our experiments. All Qwen3 models refer to their Instruct versions.

| Model | Total | Active | Experts | Top-$k$ |
| --- | --- | --- | --- | --- |
| Qwen3-30B-A3B | 30B | 3B | 128 | 8 |
| GPT-OSS-20B | 21B | 3.6B | 32 | 4 |
| Ling-Lite-1.5 | 16B | 3B | 64 | 6 |
| Qwen3-Next-80B-A3B | 80B | 3B | 256 | 10 |

**Datasets.** To comprehensively evaluate the generalization and effectiveness of Expert-Sample, we conduct experiments on multiple benchmarks. For scaling experiments (Section 4.2), we evaluate on AIME-120, GPQA-Diamond, and LiveCodeBench-V6-Lite (Jain et al., 2024), covering mathematics, knowledge reasoning, and code generation to validate scaling efficiency across diverse task types.

For verification experiments (Section 4.3), we focus on a broader range of specific tasks for generalization: AIME 2024, AIME 2025, MATH-500-Hard (Level-5), HMMT 2025, and GPQA-Diamond. For benchmarks with fewer samples (AIME 2024-2025 and HMMT 2025), we run 5 independent trials and report the average to reduce variance.

**Implementation Details.** For token-level sampling, we use two temperature configurations: (1) *normal temperature*: $T = 0.7$, top-$p = 0.8$, top-$k = 20$, following the official model card recommendations; and (2) *high temperature*: $T = 1.3$, top-$p = 0.98$, top-$k =$ None, designed to maximize diversity. Following Section 3.4, we adopt a unified hyperparameter setting for Expert-Sample across all models: $k_{\text{keep}} = \lfloor k/2 \rfloor + 1$, $\tau = 1.0$, and $r = 4k$, ($k$ is the default number of activated experts). Expert-Sample is combined with normal-temperature token sampling in all experiments. We use LightEval (Habib et al., 2023) as evaluatin framework and vLLM (Kwon et al., 2023) as inference backend.

### 4.2. Scaling Experiments

In this section, we evaluate how pass@n accuracy scales with increasing computational budget (measured by the number of generated samples $n$) when combining Expert-Sample with four fine-grained MoE models across three task categories: mathematical reasoning, knowledge reasoning, and code generation. We compare against three baselines: normal-temperature token sampling, high-temperature token sampling, and Entropy-based Dynamic Temperature (EDT) Sampling (Zhang et al., 2024), which dynamically adjusts the temperature parameter based on token entropy.

Figure 4 presents the scaling curves on AIME-120, GPQA-Diamond, and LiveCodeBench-V6-Lite. As can be seen:

High-temperature token sampling generally improves pass@n accuracy compared to normal-temperature sampling, but the gains are limited and inconsistent—in several cases, it actually underperforms normal-temperature sampling, revealing its inherent instability. EDT provides a more principled approach by dynamically adjusting temperature based on token entropy, achieving improvements over fixed-temperature baselines in most cases. However, EDT's gains remain modest, as it still operates at the token level and thus faces the fundamental stability-diversity trade-off.

In contrast, Expert-Sample delivers stable and substantial improvements across all experimental conditions. Notably,

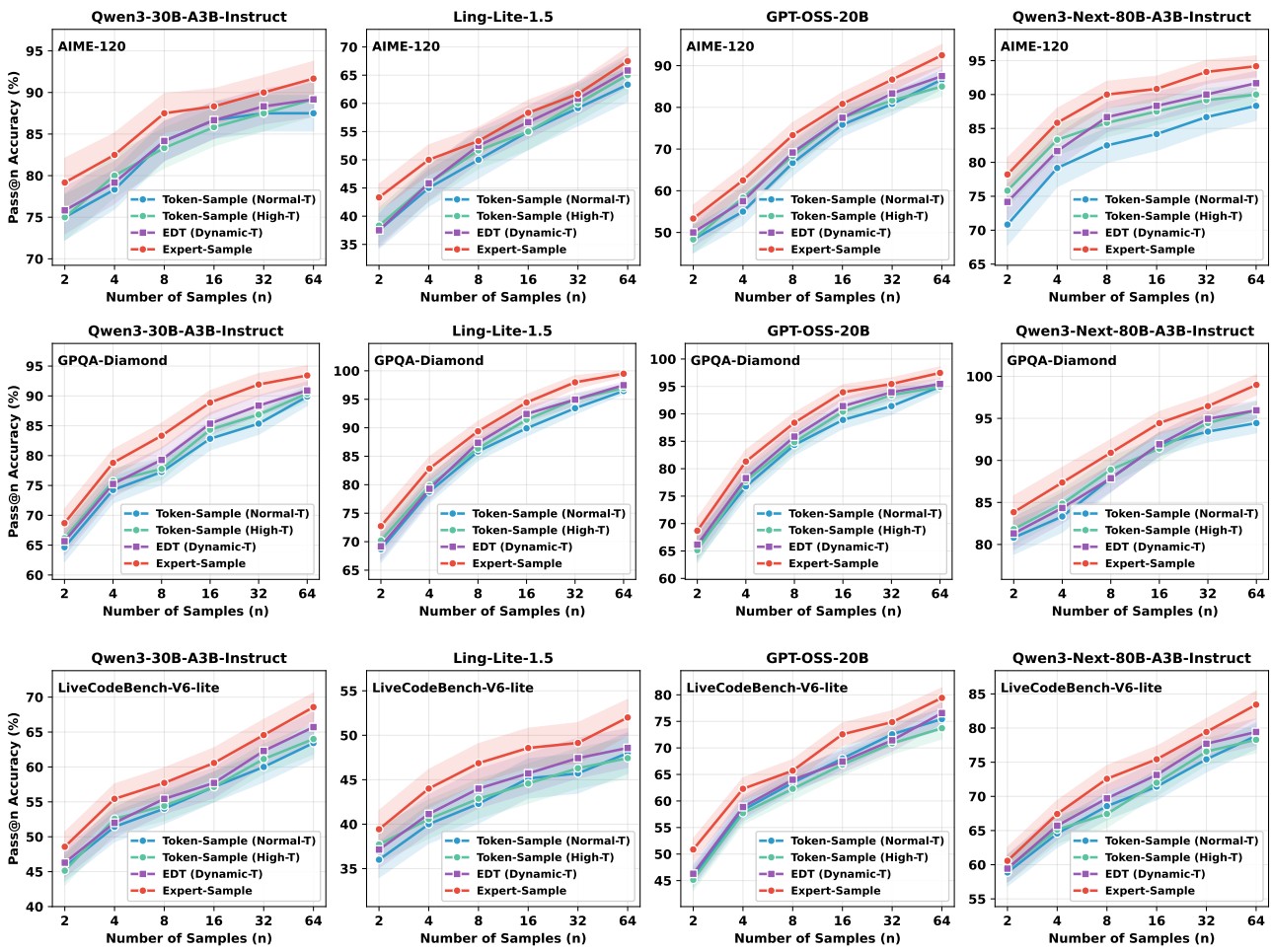

*Figure 4.* Pass@n accuracy scaling curves on AIME-120 (top), GPQA-Diamond (middle), and LiveCodeBench-V6-Lite (bottom). Shaded regions indicate standard deviation. Expert-Sample consistently outperforms token sampling baselines across all models and tasks.

Expert-Sample brings gains even at low sample counts, and this advantage persists rather than diminishes as the number of samples increases. At pass@64, Expert-Sample yields an average improvement of 4.32% over normal-temperature token sampling across all 12 model-benchmark combinations. On GPQA-Diamond, a challenging graduate-level knowledge reasoning benchmark, Expert-Sample further pushes all four models to near-perfect pass@64 accuracy, demonstrating the method's potential on difficult tasks.

These results show that by injecting diversity at the expert routing level, Expert-Sample sidesteps the stability-diversity trade-off inherent to token-level approaches, enabling more efficient exploration of the solution space.

### 4.3. Verification Experiments

In the previous section, we demonstrated that Expert-Sample significantly improves the probability of finding correct solutions as the number of samples increases. In test-time scaling, verification is equally crucial, as a ro-

bust verification pipeline can directly select the final output from multiple candidates. In this section, we combine Expert-Sample with commonly used verification methods to demonstrate that our approach improves actual accuracy in practical scenarios. We evaluate two verification methods:

**Best-of-N (BoN).** This method leverages an external reward model to score each response and selects the highest-scoring response as final answer. We use Qwen2.5-Math-PRM-7B and Llama3.1-8B-PRM-Deepseek-Data as reward models.

**Weighted Majority Voting (WMV).** This method first scores each response using the reward model, then groups responses by their final answers and computes the average score within each group. The answer with the highest average group score is selected as the final output, leveraging both the frequency signal from multiple samples and the quality signal from the reward model.

Notably, Expert-Sample is a fundamental sampling method that improves the diversity of model outputs and is orthogonal to any verification method. We therefore verify that

*Table 2.* Accuracy (%) comparison of different verification methods with and without Expert-Sample (+ES) across four models and five benchmarks. Expert-Sample consistently improves accuracy when combined with both BoN and WMV verification methods.

| Model | Method | AIME2024 | AIME2025 | HMMT-25 | MATH-Hard | GPQA-Diamond |
|-------|--------|----------|----------|---------|-----------|--------------|
| Qwen3-30B-A3B-Instruct | BoN | 84.00 | 70.00 | 50.67 | 89.55 | 59.09 |
| | BoN + ES | $86.67_{\uparrow 2.67}$ | $76.33_{\uparrow 6.33}$ | $55.33_{\uparrow 4.66}$ | $91.79_{\uparrow 2.24}$ | $62.63_{\uparrow 3.54}$ |
| | WMV | 87.33 | 73.33 | 52.67 | 91.04 | 61.62 |
| | WMV + ES | $91.33_{\uparrow 4.00}$ | $76.67_{\uparrow 3.34}$ | $56.33_{\uparrow 3.66}$ | $93.28_{\uparrow 2.24}$ | $63.13_{\uparrow 1.51}$ |
| Ling-Lite-1.5 | BoN | 53.33 | 39.33 | 26.67 | 81.34 | 59.60 |
| | BoN + ES | $56.00_{\uparrow 2.67}$ | $45.00_{\uparrow 5.67}$ | $33.33_{\uparrow 6.66}$ | $84.33_{\uparrow 2.99}$ | $62.12_{\uparrow 2.52}$ |
| | WMV | 56.67 | 37.33 | 26.00 | 82.84 | 59.09 |
| | WMV + ES | $59.67_{\uparrow 3.00}$ | $40.67_{\uparrow 3.34}$ | $30.00_{\uparrow 4.00}$ | $85.07_{\uparrow 2.23}$ | $61.61_{\uparrow 2.52}$ |
| GPT-OSS-20B | BoN | 53.33 | 49.33 | 36.67 | 75.37 | 53.53 |
| | BoN + ES | $59.33_{\uparrow 6.00}$ | $55.67_{\uparrow 6.34}$ | $42.67_{\uparrow 6.00}$ | $78.36_{\uparrow 2.99}$ | $58.08_{\uparrow 4.55}$ |
| | WMV | 57.33 | 49.33 | 39.33 | 79.85 | 56.06 |
| | WMV + ES | $63.00_{\uparrow 5.67}$ | $53.67_{\uparrow 4.34}$ | $43.33_{\uparrow 4.00}$ | $82.09_{\uparrow 2.24}$ | $58.59_{\uparrow 2.53}$ |
| Qwen3-Next-80B-A3B-Instruct | BoN | 83.33 | 76.67 | 54.00 | 89.55 | 72.72 |
| | BoN + ES | $88.67_{\uparrow 5.34}$ | $80.67_{\uparrow 4.00}$ | $59.33_{\uparrow 5.33}$ | $91.04_{\uparrow 1.49}$ | $76.26_{\uparrow 3.54}$ |
| | WMV | 84.00 | 73.33 | 56.33 | 90.30 | 74.74 |
| | WMV + ES | $87.33_{\uparrow 3.33}$ | $76.33_{\uparrow 3.00}$ | $59.67_{\uparrow 3.34}$ | $92.54_{\uparrow 2.24}$ | $77.27_{\uparrow 2.53}$ |

combining Expert-Sample with these methods yields higher actual accuracy. For each model on each dataset, we generate 32 responses and apply verification methods to select the final answer. The baseline uses normal-temperature token sampling, while +ES denotes adding Expert-Sample on top of normal-temperature token sampling.

Table 2 presents the results across four models. Although BoN and WMV exhibit varying relative performance across different model-dataset combinations, both consistently benefit from Expert-Sample. Across 20 model-dataset combinations, Expert-Sample yields average accuracy improvements of 4.28% on top of BoN and 3.15% on top of WMV.

These results demonstrate the practical utility of Expert-Sample: it serves as a complementary technique that enhances the effectiveness of existing verification methods by providing more diverse candidate responses for selection.

## 5. Overhead Analysis

Although Expert-Sample introduces additional operations during expert selection, these operations are computationally lightweight and fully vectorized, and expert selection constitutes only a tiny fraction of the total inference cost. Consequently, the per-path overhead is negligible in practice. We emphasize that the analysis here concerns the additional cost introduced by Expert-Sample on each individual generation path; the total cost of test-time scaling naturally scales with the number of candidate paths generated, which is orthogonal to our method.

To validate this, we measured end-to-end latency on Qwen3-30B-A3B-Instruct and Ling-Lite-1.5 using vLLM

on 8xA800-80G GPUs with random inputs sampled from WikiText-2. Table 3 reports throughput for prefill and decode phases.

*Table 3.* Throughput (tokens/s) comparison with and without Expert-Sample. We set prompt length = 1024, batch size = 8, output length = 1024. Values in parentheses indicate relative change.

| Model | Method | Prefill | Decode |
|-------|--------|---------|--------|
| Qwen3-30B-A3B -Instruct | Baseline | 136754 | 666.7 |
| | +Expert-Sample | 137121 ($\uparrow$0.27%) | 662.1 ($\downarrow$0.70%) |
| Ling-Lite-1.5 | Baseline | 214802 | 1024 |
| | +Expert-Sample | 213463 ($\downarrow$0.62%) | 1028 ($\uparrow$0.40%) |

The relative change in per-path throughput for both prefill and decode phases remains within $\pm 1\%$, which is almost within the range of measurement noise. This confirms that Expert-Sample is lightweight and introduces negligible additional cost per generation path. We provide comprehensive results across all models and configurations in Appendix B.

## 6. Robustness of Diversity Evaluation

To further validate the reliability of our LLM-based pairwise reasoning diversity evaluation in Section 3.3, we examine both judge consistency and robustness to alternative judges. All experiments are conducted on the Uncertain Set of AIME120 on Qwen3-30B-A3B-Instruct.

**Judge Consistency.** We re-run the DeepSeek-R1 pairwise evaluation three times. As shown in Table 4, the standard deviations remain small across runs, and the relative gaps between methods are stable, indicating the robustness.

*Table 4.* Judge consistency: DeepSeek-R1 pairwise reasoning diversity scores across three independent runs.

| Method | Run 1 | Run 2 | Run 3 | Avg $\pm$ Std |
|---|---|---|---|---|
| Token-Sample ($t$=0.7) | 0.214 | 0.229 | 0.241 | $0.228 \pm 0.014$ |
| Token-Sample ($t$=1.3) | 0.295 | 0.291 | 0.303 | $0.296 \pm 0.006$ |
| Expert-Sample (Ours) | 0.371 | 0.386 | 0.401 | $0.386 \pm 0.015$ |

**Robustness to Alternative Judges.** We replace the judge with GPT-4o (OpenAI, 2024) and Claude-Sonnet-4.6 respectively and repeat each evaluation three times. Results are reported in Table 5. While different judges assign slightly different absolute scores, the relative advantage of Expert-Sample over Token-Sample remains clearly evident across all judges, confirming that our diversity findings are robust to alternative judges.

*Table 5.* Cross-judge robustness: pairwise reasoning diversity scores (Avg $\pm$ Std over three runs) with different LLM judges.

| Method | GPT-4o | Claude-Sonnet-4.6 |
|---|---|---|
| Token-Sample ($t$=0.7) | $0.294 \pm 0.007$ | $0.258 \pm 0.010$ |
| Token-Sample ($t$=1.3) | $0.349 \pm 0.008$ | $0.321 \pm 0.008$ |
| Expert-Sample (Ours) | $0.422 \pm 0.005$ | $0.403 \pm 0.012$ |

## 7. Related Work

### 7.1. Test-Time Scaling

Test-time scaling improves model performance by generating multiple candidates to cover the correct solution and selecting the correct one among them. One line of research focuses on enhancing diversity in multi-sample generation, either through dynamic temperature adjustment (Zhang et al., 2024) or by rephrasing questions (Zhou et al., 2024) and modifying prompts (Naik et al., 2024). Another line of work focuses on guiding search and verifying results, including beam search (Guo et al., 2024), tree-based reasoning (Yao et al., 2023), and learned or rule-based verifiers (Irvine et al., 2023). Overall, the effectiveness of test-time scaling depends on two factors: whether the correct answer is covered among multiple samples, and whether it can be successfully identified. We focus on the former aspect, aiming to enhance reasoning diversity to improve coverage of correct solutions. which also serves as the foundation for the latter.

### 7.2. Fine-Grained MoE

With a substantially larger pool of well-trained experts, fine-grained MoE offers greater flexibility in expert selection. Numerous works (Lu et al., 2024; Chen et al., 2025a) explore expert pruning to improve inference efficiency, showing that models can maintain performance with fewer activated experts. Others manually adjust expert selection to enhance specific capabilities, such as strengthening critical experts to boost reasoning (Wang et al., 2025; Chen

et al., 2025b). These works collectively suggest greedy top-$k$ selection learned during pretraining may not be optimal, leaving significant potential for better utilizing the large expert pool at inference time.

## 8. Conclusion

We present Expert-Sample, a simple yet effective method that introduces diversity at the expert routing level in fine-grained MoE models. Unlike traditional token-level sampling, which faces an inherent trade-off between diversity and stability, Expert-Sample injects randomness earlier in the computation—at the expert selection stage, and remains compatible with normal-temperature decoding. Extensive experiments across four fine-grained MoE models and diverse reasoning tasks demonstrate that Expert-Sample consistently improves pass@n accuracy and provides actual accuracy gains when combined with verification methods.

Our work reveals that fine-grained MoE architectures possess untapped potential for inference-time scaling through expert-level interventions. We hope this work inspires further exploration of this complementary dimension to token-level strategies, and provides insights for the training side.

**Appendix Overview.** To help readers navigate the supplementary material without missing key information, we summarize the most important appendix contents here:

1. Implementation details of Expert-Sample (Appendix A).

2. Detailed overhead analysis (Appendix B).

3. Hyperparameter sensitivity analysis (Appendix C).

4. LLM-based diversity evaluation details (Appendix D).

5. Extended router weight distribution analysis across models and datasets (Appendix E).

## Acknowledgements

This work is supported by the Zhongguancun Academy, (Grant No.s C20250505). This work is supported by the National Natural Science Foundation of China (NO.62572471) and the General Research Fund (GRF 16209124).

## Impact Statement

This paper presents work whose goal is to advance the field of Machine Learning, specifically improving inference-time scaling for Mixture-of-Experts models. Our method, Expert-Sample, is a general-purpose sampling technique that enhances reasoning diversity without additional computational overhead. We do not foresee any direct negative societal consequences arising from this work.

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

---

**Algorithm 1** Expert-Sample

---

**Require:** Router logits $\mathbf{L} \in \mathbb{R}^{N \times E}$ where $N$ is the number of tokens and $E$ is the number of experts, number of experts to activate $k$, number of experts to keep $k_{\text{keep}}$, sampling range $r$, temperature $\tau$

**Ensure:** Selected expert indices $\mathbf{S} \in \mathbb{Z}^{N \times k}$, corresponding weights $\mathbf{W} \in \mathbb{R}^{N \times k}$

1: $\mathbf{\Pi} \leftarrow \text{argsort}(\mathbf{L}, \text{descending})$       ▷ Sort experts by logits, $\mathbf{\Pi} \in \mathbb{Z}^{N \times E}$
2: $\mathbf{S}_{\text{keep}} \leftarrow \mathbf{\Pi}[:, 1 : k_{\text{keep}}]$       ▷ Keep top-$k_{\text{keep}}$ experts
3: $\mathbf{C} \leftarrow \mathbf{\Pi}[:, k_{\text{keep}} + 1 : r]$       ▷ Candidate pool for sampling
4: $\tilde{\mathbf{L}} \leftarrow \text{gather}(\mathbf{L}, \mathbf{C})/\tau$       ▷ Temperature-scaled candidate logits
5: $\mathbf{G} \leftarrow -\log(-\log(\mathbf{U}))$, where $\mathbf{U} \sim \text{Uniform}(0, 1)^{N \times (r - k_{\text{keep}})}$       ▷ Gumbel noise
6: $\mathbf{S}_{\text{sample}} \leftarrow \text{gather}(\mathbf{C}, \text{argtop-}(k - k_{\text{keep}})(\tilde{\mathbf{L}} + \mathbf{G}))$       ▷ Sample via Gumbel-Top-K
7: $\mathbf{S} \leftarrow \text{concat}(\mathbf{S}_{\text{keep}}, \mathbf{S}_{\text{sample}})$
8: $\mathbf{W} \leftarrow \text{gather}(\text{softmax}(\mathbf{L}), \mathbf{S})$       ▷ Compute weights from original logits
9: $\mathbf{W} \leftarrow \mathbf{W}/\text{sum}(\mathbf{W}, \text{dim} = -1)$       ▷ Renormalize weights
10: **return** $\mathbf{S}, \mathbf{W}$

---

# A. Implementation Details of Expert-Sample

## A.1. Gumbel-Top-K Sampling

The Gumbel-Top-K trick provides an efficient method for sampling $k$ elements from a categorical distribution without replacement. Given a set of unnormalized log-probabilities (logits) $\{\ell_1, \ell_2, \ldots, \ell_n\}$, the algorithm adds independent Gumbel noise to each logit and selects the top-$k$ perturbed values:

$$S = \underset{|S|=k}{\arg\max} \sum_{i \in S} (\ell_i + G_i), \quad G_i \sim \text{Gumbel}(0, 1) \tag{5}$$

where the Gumbel noise is generated as $G_i = -\log(-\log(U_i))$ with $U_i \sim \text{Uniform}(0, 1)$.

This approach offers two key advantages. First, it is algorithmically equivalent to sampling without replacement, where the probability of selecting each element is proportional to its exponentiated logit (i.e., its softmax weight). Second, it is computationally efficient: the entire sampling procedure reduces to a single top-$k$ operation over the perturbed logits, which can be executed in $O(n + k \log k)$ time and is highly parallelizable on modern hardware.

## A.2. Algorithm

Algorithm 1 presents the pseudocode for Expert-Sample. The procedure first retains the top-$k_{\text{keep}}$ experts with the highest router scores, then samples the remaining $k - k_{\text{keep}}$ experts from the subsequent candidates using the Gumbel-Top-K trick with optional temperature scaling.

## A.3. Implementation in vLLM

We implement Expert-Sample by modifying the expert selection function in vLLM. The original implementation uses deterministic top-$k$ selection:

*Listing 1.* Original expert selection in vLLM.

```python
@staticmethod
def select_experts(
    hidden_states: torch.Tensor,
    router_logits: torch.Tensor,
    top_k: int,
    renormalize: bool,
    scoring_func: str = "softmax",
    indices_type: Optional[torch.dtype] = None,
    # other arguments omitted for brevity
) -> tuple[torch.Tensor, torch.Tensor]:
```

```python
    """Simplified expert selection using deterministic top-k."""

    assert hidden_states.size(0) == router_logits.size(0), (
        "Number of tokens mismatch")
    if scoring_func != "softmax":
        raise ValueError("Only softmax scoring function is supported in
            select_experts_simple.")

    # Compute softmax probabilities over experts
    scores = torch.softmax(router_logits, dim=-1).to(torch.float32)

    # Top-k selection per token
    topk_weights, topk_ids = torch.topk(
        scores, k=top_k, dim=-1, sorted=False)

    # Renormalize weights across selected experts
    if renormalize:
        topk_weights = topk_weights / topk_weights.sum(
            dim=-1, keepdim=True)

    if indices_type is not None:
        topk_ids = topk_ids.to(dtype=indices_type)

    return topk_weights, topk_ids
```

Our modified implementation introduces the Expert-Sample mechanism:

*Listing 2.* Expert-Sample implementation.

```python
@staticmethod
def select_experts_sample(
    hidden_states: torch.Tensor,
    router_logits: torch.Tensor,
    top_k: int,
    renormalize: bool,
    k_keep: int,              # 0 < k_keep < top_k
    sample_scale: int,        # top_k < sample_scale <= num_experts
    temperature: float,       # temperature > 0
    scoring_func: str = "softmax",
    indices_type: Optional[torch.dtype] = None,
    # other arguments omitted for brevity
) -> tuple[torch.Tensor, torch.Tensor]:
    """Expert selection with Gumbel-Top-K sampling."""

    batch_size, num_experts = router_logits.shape
    n_sample = top_k - k_keep

    # Sort experts by logits in descending order
    _, idx_desc = torch.sort(
        router_logits, dim=-1, descending=True)

    # Step 1: Keep top k_keep experts
    keep_ids = idx_desc[:, :k_keep]

    # Step 2: Sample from candidate pool (k_keep, sample_scale]
```

```
candidate_indices = idx_desc[:, k_keep:sample_scale]
batch_indices = torch.arange(
    batch_size, device=router_logits.device
).unsqueeze(-1).expand_as(candidate_indices)
candidate_logits = router_logits[batch_indices, candidate_indices]

# Step 3: Gumbel-Top-K sampling with temperature scaling
candidate_logits = candidate_logits / temperature
gumbel_noise = -torch.log(-torch.log(
    torch.rand_like(candidate_logits) + 1e-10) + 1e-10)

perturbed_logits = candidate_logits + gumbel_noise
_, sample_indices = torch.topk(
    perturbed_logits, k=n_sample, dim=-1, sorted=False)
sampled_ids = candidate_indices.gather(-1, sample_indices)

# Step 5: Combine kept and sampled experts
topk_ids = torch.cat([keep_ids, sampled_ids], dim=-1)

# Compute weights from original logits and renormalize
scores = torch.softmax(router_logits, dim=-1).to(torch.float32)
topk_weights = scores.gather(-1, topk_ids)

if renormalize:
    topk_weights = topk_weights / topk_weights.sum(
        dim=-1, keepdim=True)

if indices_type is not None:
    topk_ids = topk_ids.to(dtype=indices_type)

return topk_weights, topk_ids
```

## B. Overhead Analysis of Expert-Sample

Although Expert-Sample introduces additional operations during expert selection, the overhead is negligible in practice. This can be attributed to two main factors. First, all introduced operations are vectorized and computationally lightweight, consisting primarily of sorting, Gumbel noise generation, and top-$k$ selection. Thanks to the efficient implementation of Gumbel-Top-K sampling, which requires only element-wise random number generation and a single top-$k$ operation, these computations are highly optimized on modern GPUs. Second, expert selection constitutes only a tiny fraction of the total end-to-end computation in MoE models. The dominant computational costs lie in the attention mechanism and the expert FFN computations, while the router network is merely a small linear projection followed by a softmax. Consequently, even if the expert selection overhead increases moderately, its impact on overall latency remains within measurement noise due to the negligible proportion of routing in the total computation.

To empirically validate this analysis, we conducted end-to-end latency experiments on all four fine-grained MoE models used in this paper. We separately measured the prefill phase and decode phase latencies, comparing the original implementation against our Expert-Sample variant. All experiments were conducted using vLLM as the inference backbone on $8\times$ A800-80G GPUs with tensor parallelism set to 8. The prompts used in all experiments were randomly sampled from the WikiText-2 dataset (Merity et al., 2016).

### B.1. Prefill Phase Overhead

For the prefill phase, we varied the prompt length across 256, 512, and 1024 tokens, and the batch size across 1, 2, 4, 8, 16, and 32. Table 6 reports the throughput in tokens per second, with the relative ratio to the baseline shown in parentheses.

*Table 6.* Prefill phase throughput (tokens/s) comparison between baseline and Expert-Sample. Values in parentheses indicate relative ratio.

| Model | Method | BS=1 | BS=2 | BS=4 | BS=8 | BS=16 | BS=32 |
|---|---|---|---|---|---|---|---|
| | | *Prompt Length = 256* | | | | | |
| Qwen3-30B-A3B-Instruct | Baseline | 16382 | 18742 | 31049 | 45578 | 81998 | 93531 |
| | +Expert-Sample | 16397 (100.09%) | 18736 (99.97%) | 30950 (99.68%) | 45753 (100.38%) | 82282 (100.35%) | 94157 (100.67%) |
| Ling-lite-1.5 | Baseline | 24355 | 28520 | 54191 | 98390 | 124885 | 166057 |
| | +Expert-Sample | 24424 (100.29%) | 28053 (98.36%) | 53798 (99.27%) | 96900 (98.49%) | 124248 (99.49%) | 165758 (99.82%) |
| GPT-OSS-20B | Baseline | 22923 | 28593 | 45638 | 77378 | 102119 | 145676 |
| | +Expert-Sample | 22769 (99.33%) | 28487 (99.63%) | 45635 (99.99%) | 77337 (99.95%) | 103140 (101.00%) | 145775 (100.07%) |
| Qwen3-Next-80B-A3B-Instruct | Baseline | 3031 | 3120 | 5236 | 8079 | 10914 | 12943 |
| | +Expert-Sample | 3057 (100.86%) | 3142 (100.71%) | 5247 (100.20%) | 8031 (99.41%) | 10736 (98.37%) | 12760 (98.59%) |
| | | *Prompt Length = 512* | | | | | |
| Qwen3-30B-A3B-Instruct | Baseline | 32358 | 35459 | 59847 | 89556 | 107253 | 179978 |
| | +Expert-Sample | 32394 (100.11%) | 35447 (99.97%) | 59842 (99.99%) | 89792 (100.26%) | 106181 (99.00%) | 180650 (100.37%) |
| Ling-lite-1.5 | Baseline | 41438 | 48030 | 88540 | 106763 | 180304 | 250915 |
| | +Expert-Sample | 40810 (98.49%) | 47122 (98.11%) | 87986 (99.37%) | 110792 (103.77%) | 179279 (99.43%) | 254726 (101.52%) |
| GPT-OSS-20B | Baseline | 52264 | 56495 | 89853 | 149179 | 201865 | 282055 |
| | +Expert-Sample | 52034 (99.56%) | 56359 (99.76%) | 89747 (99.88%) | 147687 (99.00%) | 200774 (99.46%) | 280789 (99.55%) |
| Qwen3-Next-80B-A3B-Instruct | Baseline | 5241 | 5343 | 8074 | 10929 | 13108 | 14517 |
| | +Expert-Sample | 5236 (99.91%) | 5312 (99.43%) | 8116 (100.52%) | 10826 (99.06%) | 12883 (98.29%) | 14243 (98.11%) |
| | | *Prompt Length = 1024* | | | | | |
| Qwen3-30B-A3B-Instruct | Baseline | 58281 | 65590 | 107863 | 136754 | 203455 | 335339 |
| | +Expert-Sample | 58350 (100.12%) | 65566 (99.96%) | 108256 (100.36%) | 137121 (100.27%) | 204264 (100.40%) | 338692 (101.00%) |
| Ling-lite-1.5 | Baseline | 72607 | 88013 | 159705 | 214802 | 281129 | 408731 |
| | +Expert-Sample | 71976 (99.13%) | 86711 (98.52%) | 160963 (100.79%) | 213463 (99.38%) | 279516 (99.43%) | 410366 (100.40%) |
| GPT-OSS-20B | Baseline | 80361 | 94136 | 157069 | 196456 | 273379 | 377710 |
| | +Expert-Sample | 80420 (100.07%) | 94232 (100.10%) | 157156 (100.06%) | 196661 (100.10%) | 275396 (100.74%) | 380567 (100.76%) |
| Qwen3-Next-80B-A3B-Instruct | Baseline | 8057 | 8189 | 11061 | 13253 | 14556 | 14843 |
| | +Expert-Sample | 8011 (99.43%) | 8175 (99.83%) | 10841 (98.01%) | 13168 (99.36%) | 14351 (98.60%) | 14460 (97.42%) |

*Table 7.* Decode phase throughput (tokens/s) comparison between baseline and Expert-Sample. Prompt length is fixed at 1024.

| Model | Method | BS=1 | BS=2 | BS=4 | BS=8 | BS=16 | BS=32 |
|---|---|---|---|---|---|---|---|
| | | *Output Length = 256* | | | | | |
| Qwen3-30B-A3B-Instruct | Baseline | 105.6 | 194.3 | 374.2 | 675.5 | 1262 | 1993 |
| | +Expert-Sample | 106.7 (101.00%) | 195.5 (100.60%) | 375.5 (100.34%) | 675.7 (100.03%) | 1260 (99.80%) | 2007 (100.70%) |
| Ling-lite-1.5 | Baseline | 168.0 | 304.4 | 572.4 | 1052 | 1931 | 3136 |
| | +Expert-Sample | 168.3 (100.17%) | 303.5 (99.71%) | 582.4 (101.74%) | 1066 (101.26%) | 1854 (96.03%) | 3144 (100.23%) |
| GPT-OSS-20B | Baseline | 170.0 | 293.0 | 565.7 | 1020 | 1888 | 2946 |
| | +Expert-Sample | 168.4 (99.04%) | 295.9 (101.00%) | 561.5 (99.26%) | 1015 (99.50%) | 1873 (99.20%) | 2948 (100.06%) |
| Qwen3-Next-80B-A3B-Instruct | Baseline | 103.0 | 168.3 | 329.4 | 603.1 | 1119 | 1812 |
| | +Expert-Sample | 103.1 (100.05%) | 167.2 (99.38%) | 323.6 (98.22%) | 586.7 (97.28%) | 1116 (99.73%) | 1824 (100.67%) |
| | | *Output Length = 512* | | | | | |
| Qwen3-30B-A3B-Instruct | Baseline | 105.4 | 193.5 | 373.7 | 672.5 | 1254 | 1990 |
| | +Expert-Sample | 105.7 (100.30%) | 194.5 (100.54%) | 374.2 (100.12%) | 668.7 (99.44%) | 1242 (99.00%) | 2004 (100.72%) |
| Ling-lite-1.5 | Baseline | 164.4 | 298.6 | 571.1 | 1049 | 1901 | 3071 |
| | +Expert-Sample | 165.4 (100.59%) | 299.3 (100.21%) | 574.0 (100.51%) | 1052 (100.31%) | 1905 (100.16%) | 3115 (101.44%) |
| GPT-OSS-20B | Baseline | 169.3 | 292.2 | 562.6 | 1016 | 1877 | 2907 |
| | +Expert-Sample | 167.6 (99.00%) | 290.3 (99.36%) | 559.4 (99.43%) | 1012 (99.56%) | 1863 (99.25%) | 2919 (100.41%) |
| Qwen3-Next-80B-A3B-Instruct | Baseline | 102.7 | 167.9 | 328.5 | 603.8 | 1110 | 1801 |
| | +Expert-Sample | 99.7 (97.04%) | 167.4 (99.69%) | 325.4 (99.05%) | 602.7 (99.82%) | 1095 (98.67%) | 1797 (99.78%) |
| | | *Output Length = 1024* | | | | | |
| Qwen3-30B-A3B-Instruct | Baseline | 105.1 | 192.5 | 372.9 | 666.7 | 1243 | 1971 |
| | +Expert-Sample | 106.1 (100.97%) | 193.6 (100.56%) | 373.1 (100.04%) | 662.1 (99.30%) | 1233 (99.20%) | 1991 (101.00%) |
| Ling-lite-1.5 | Baseline | 160.0 | 288.8 | 555.2 | 1024 | 1860 | 3037 |
| | +Expert-Sample | 160.6 (100.37%) | 291.3 (100.86%) | 540.5 (97.35%) | 1028 (100.40%) | 1814 (97.52%) | 3031 (99.83%) |
| GPT-OSS-20B | Baseline | 167.3 | 291.6 | 560.2 | 1011 | 1858 | 2888 |
| | +Expert-Sample | 166.2 (99.36%) | 292.2 (100.21%) | 560.7 (100.08%) | 1008 (99.65%) | 1852 (99.70%) | 2892 (100.15%) |
| Qwen3-Next-80B-A3B-Instruct | Baseline | 102.6 | 167.8 | 327.5 | 602.1 | 1109 | 1799 |
| | +Expert-Sample | 100.3 (97.77%) | 162.5 (96.81%) | 326.3 (99.64%) | 599.8 (99.62%) | 1108 (99.95%) | 1788 (99.42%) |

## B.2. Decode Phase Overhead

For the decode phase, we fixed the prompt length at 1024 tokens and varied the output length across 256, 512, and 1024 tokens, with batch sizes ranging from 1 to 32. Table 7 reports the throughput in tokens per second.

As shown in Tables 6 and 7, the throughput ratios remain consistently close to 100% across all configurations, with most values falling within the range of 97%–103%. These minor variations are well within the range of measurement noise, confirming that Expert-Sample introduces negligible overhead. This empirical evidence validates our theoretical analysis that the expert selection component contributes minimally to the overall computational cost of MoE inference.

## C. Hyperparameter Analysis

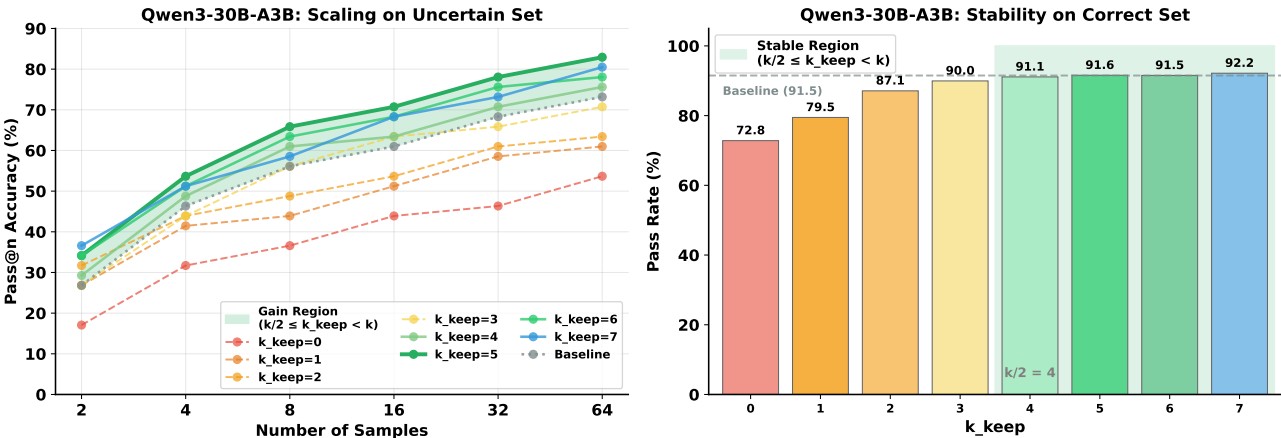

*Figure 5.* Effect of $k_{\text{keep}}$ on Qwen3-30B-A3B (Instruct). **Left:** Pass@$n$ accuracy on the Uncertain Set across different $k_{\text{keep}}$ values. The shaded region indicates the recommended range ($k/2 \leq k_{\text{keep}} < k$) where Expert-Sample consistently outperforms the baseline. **Right:** Pass rate on the Correct Set. Stability degrades significantly when $k_{\text{keep}} < k/2$, but remains comparable to baseline once $k_{\text{keep}} \geq k/2$.

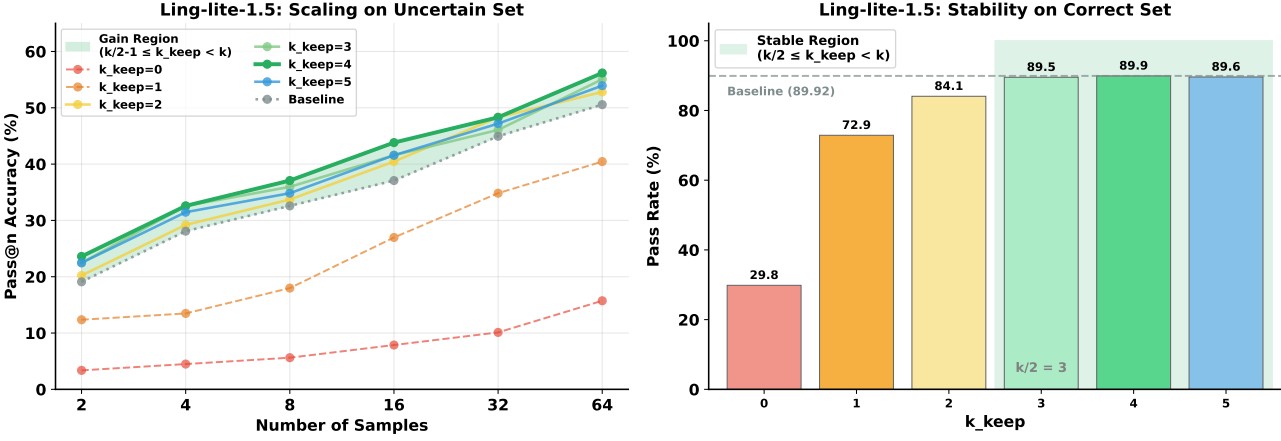

*Figure 6.* Effect of $k_{\text{keep}}$ on Ling-Lite-1.5. Similar patterns emerge: the stable and effective region lies within $k/2 \leq k_{\text{keep}} < k$, demonstrating that the recommended setting generalizes across different MoE architectures.

As discussed in Section 3.4, Expert-Sample introduces three hyperparameters: $k_{\text{keep}}$ for stability preservation, temperature $\tau$ for controlling sampling randomness, and sampling range $r$ for determining the candidate pool size. In this appendix, we provide comprehensive experimental analysis of how each hyperparameter affects Expert-Sample's performance, validating our claim that the method is robust to hyperparameter choices and offering practical guidelines for practitioners.

We organize this analysis in a Q&A format, addressing each hyperparameter in turn. Following the evaluation framework established in the main text, we assess performance along two dimensions: *stability* (measured by pass rate on the Correct

Set) and *diversity/exploration* (measured by pass@$n$ scaling on the Uncertain Set, which combines the Medium and Hard Sets). All experiments are conducted on AIME-120 using Qwen3-30B-A3B-Instruct and Ling-Lite-1.5 as testbeds.

### C.1. Is $k_{\text{keep}}$ necessary? If so, how to set it simply and effectively?

The motivation for introducing $k_{\text{keep}}$ aligns directly with our observation in Section 2.1: under greedy token decoding, preserving the selection of top-weighted experts maintains the model's accuracy and core capabilities, and empirically, keeping approximately half or slightly more than half of the experts suffices to achieve this. To validate this hypothesis, we vary $k_{\text{keep}}$ across the range $[0, k-1]$ (where $k$ is the default number of selected experts) while holding $\tau = 1.0$ and $r = 4k$ constant. For each configuration, we record the 32-run pass rate on the Correct Set to evaluate stability, and the pass@$n$ accuracy on the Uncertain Set (for $n \in \{2, 4, 8, 16, 32, 64\}$) to evaluate exploration capability.

**Results.** As shown in Figure 5 and Figure 6, the experimental results align well with our observations from Section 2.1. When $k_{\text{keep}}$ is very small or significantly less than $k/2$, the model exhibits poor stability on the Correct Set and also fails to achieve good results on the Uncertain Set. This is primarily because the top-weighted experts are crucial for the MoE model to correctly process the corresponding input tokens—they possess a degree of irreplaceability. When $k_{\text{keep}} \geq k/2$ and approaches $k$, the model maintains strong stability on the Correct Set with only minor variations. Meanwhile, on the Uncertain Set, as $k_{\text{keep}}$ approaches $k$ within this range, the behavior gradually converges to the original implementation, causing the scaling curve to approach the baseline token sampling curve. The optimal performance is typically achieved when $k_{\text{keep}}$ is slightly greater than $k/2$ but still less than $k$, and importantly, performance within this range is not highly sensitive to the exact choice of $k_{\text{keep}}$.

**Conclusion.** The $k_{\text{keep}}$ parameter is essential for Expert-Sample—without any kept experts, both stability and exploration suffer. However, as long as extreme values are avoided, Expert-Sample is robust to the choice of $k_{\text{keep}}$ while still achieving significant gains. In practice, we recommend simply setting $k_{\text{keep}} = \lfloor k/2 \rfloor + 1$ or $\lfloor k/2 \rfloor + 2$, which provides a good balance between stability and diversity without requiring task-specific tuning.

### C.2. How does temperature $\tau$ affect performance? Can high temperature balance stability and diversity?

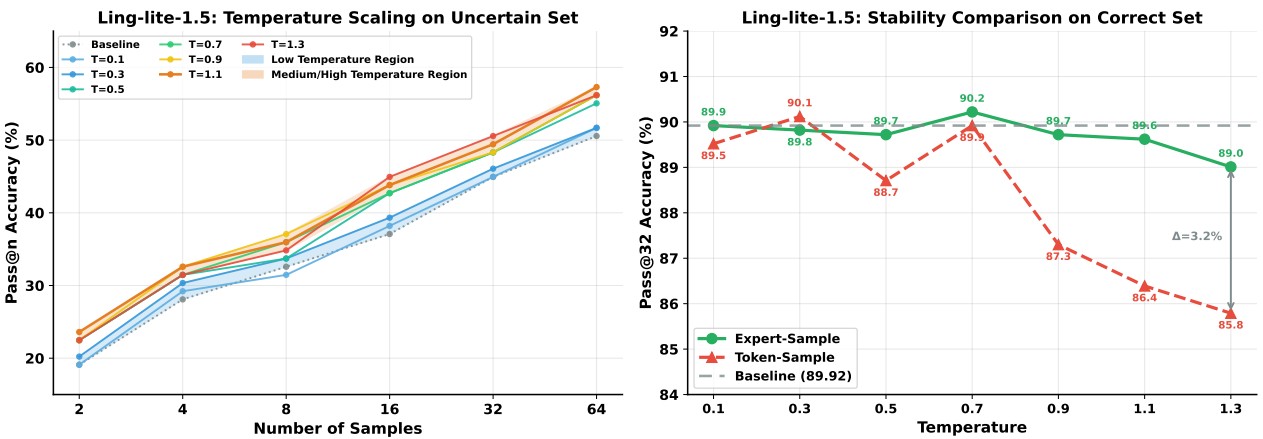

*Figure 7.* Effect of temperature $\tau$ on Ling-Lite-1.5. **Left:** Pass@$n$ accuracy on the Uncertain Set. Medium-to-high temperatures ($\tau \geq 0.7$) consistently outperform low temperatures. **Right:** Pass rate comparison on the Correct Set between Expert-Sample and Token-Sample. Expert-Sample maintains stable performance across all temperatures, while Token-Sample suffers significant stability degradation at higher temperatures (up to 3.2% gap).

As discussed earlier, one limitation of Token-Sample is the difficulty in balancing stability and diversity—using higher temperatures to encourage sampling diversity often comes at the cost of stability. To investigate whether Expert-Sample overcomes this limitation, we vary temperature $\tau$ across the range $[0.1, 1.3]$ with an interval of 0.2, covering low, medium, and high temperature settings. We use Ling-Lite-1.5 as the primary example. Following our previous analysis, we set $k_{\text{keep}} = \lfloor k/2 \rfloor + 1$ and $r = 4k$.

**Results.** As shown in Figure 7, Expert-Sample demonstrates a clear advantage in stability on the Correct Set. At low or medium temperatures, there is virtually no impact on stability; even at high temperatures, the pass rate decreases by at

most 0.5%, which is well within acceptable limits. The fundamental reason is that Expert-Sample's stability is guaranteed by $k_{\text{keep}}$—the stochastic selection of subsequent experts does not significantly affect the model's core capabilities. For comparison, we also evaluate standard token sampling under the same temperature settings (0.1, 0.3, 0.5, 0.7, 0.9, 1.1, 1.3). As temperature increases, the stability of token sampling degrades noticeably, with up to 3.2% performance gap at high temperatures. This contrast highlights the stability advantage of Expert-Sample.

On the Uncertain Set, we observe a clear scaling pattern: medium-to-high temperature Expert-Sample yields significantly higher pass@$n$ gains compared to low temperature settings. However, within the medium-to-high temperature region, pass@$n$ accuracy is not highly sensitive to temperature adjustments—the variations are relatively small. We attribute this to the fact that the lower-weighted experts have relatively similar weight distributions, and high-temperature sampling merely further reduces the weight differences among them. We consider this a positive property, as it means practitioners do not need to spend excessive effort searching for the optimal $\tau$. In practice, we recommend simply setting $\tau = 1.0$ or $\tau = 1.1$ to achieve significant gains.

### C.3. Is expanding the sampling range $r$ beneficial? How should it be set in practice?

The range parameter $r$ serves a similar role to the top-$k$ hyperparameter in token sampling, limiting the candidate pool for stochastic selection. One drawback of top-$k$ in token sampling is that users often lack clear guidance on how to set this hyperparameter. Here, we empirically explore how $r$ affects model diversity and stability, and provide practical guidelines for its configuration.

We vary $r$ from $2k$ to $7k$ with an interval of $k$, recording both the pass rate on the Correct Set and pass@$n$ on the Uncertain Set. As shown in Figure 8, on the Correct Set (right panel), the pass rate remains consistently high across all tested ranges, fluctuating only slightly between 89.21% and 89.92%—all close to the baseline of 89.92%. This demonstrates that expanding the sampling range does not compromise stability on problems the model can already solve reliably. On the Uncertain Set (left panel), as $r$ increases, the model's ability to explore difficult problems gradually improves, yielding noticeable gains. However, beyond $r = 4k$, performance essentially stabilizes with no significant further improvement, though occasional outliers may appear when $r$ becomes excessively large. In practice, we recommend setting $r = 3k$ or $r = 4k$ to balance diversity gains with stable performance.

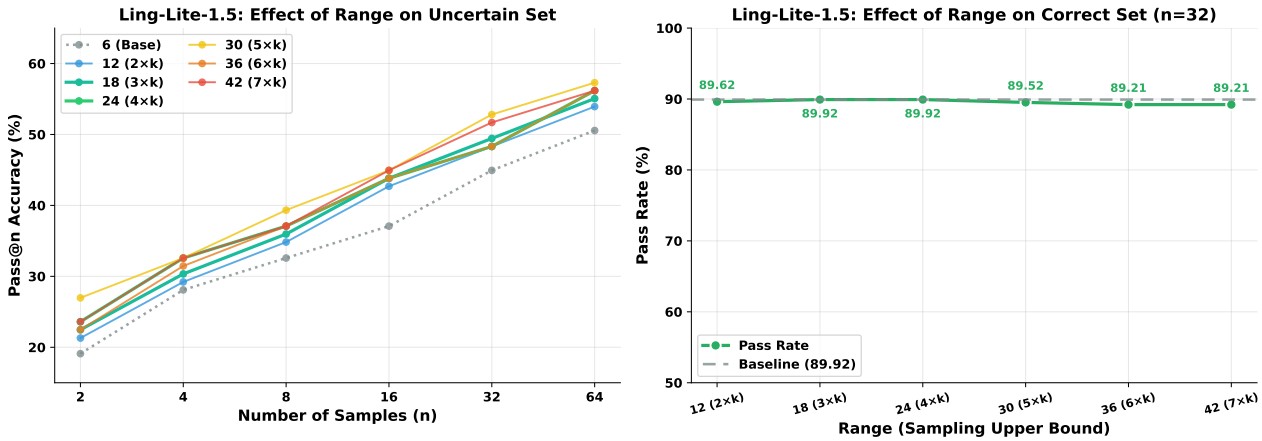

*Figure 8.* Effect of sampling range $r$ on Ling-Lite-1.5. Left: Pass@$n$ accuracy on the Uncertain Set improves as $r$ increases but stabilizes beyond $r = 4k$. Right: Pass rate on the Correct Set remains robust across all tested ranges, staying close to the baseline.

### C.4. Summary and Practical Guidelines

Our extensive hyperparameter analysis reveals that Expert-Sample is robust across a wide range of settings, requiring no task-specific tuning. We summarize the key findings and provide a simple, effective configuration for practitioners:

**Key Findings:**

- $k_{\text{keep}}$ **is essential but not sensitive.** Setting $k_{\text{keep}}$ too small (below $k/2$) harms both stability and exploration. However, within the range $[k/2, k)$, performance remains strong and stable. Simply setting $k_{\text{keep}} = \lfloor k/2 \rfloor + 1$ works reliably.

- **Temperature $\tau$ enables high diversity without sacrificing stability.** Unlike token sampling, Expert-Sample maintains stability even at high temperatures thanks to the $k_{\text{keep}}$ mechanism. Medium-to-high temperatures ($\tau \geq 0.7$) yield the best exploration gains, and performance is not highly sensitive within this range.

- **Sampling range $r$ improves diversity with diminishing returns.** Expanding $r$ enhances exploration capability, but gains stabilize beyond $r = 4k$. Setting $r = 3k$ or $4k$ provides a good balance.

**Recommended Default Configuration:**

$$k_{\text{keep}} = \lfloor k/2 \rfloor + 1, \quad \tau = 1.0, \quad r = 4k \tag{6}$$

This configuration can be directly applied to any fine-grained MoE model as a drop-in enhancement, requiring no additional tuning. For users seeking slightly more aggressive exploration, $\tau = 1.1$ and $r = 5k$ are also safe choices with minimal impact on stability.

## D. Process Diversity Evaluation Details

In this appendix, we provide details on how we use an LLM to quantify reasoning process diversity across multiple sampled responses.

### D.1. Evaluation Procedure

To measure the diversity of reasoning paths, we employ DeepSeek-R1 as a judge to evaluate the pairwise similarity between all responses generated for each problem. The evaluation follows a structured four-step process:

1. **Extract core reasoning steps from the first response**: The judge summarizes and identifies the main reasoning steps and key points from the first reasoning process.

2. **Extract core reasoning steps from the second response**: Similarly, the judge extracts the main reasoning steps from the second reasoning process.

3. **Comparative analysis**: The judge compares the two reasoning processes, focusing on whether the core approaches are the same and the degree of similarity in reasoning steps.

4. **Assign similarity score**: Based on the comparison, the judge assigns an integer score from 0 to 5, where higher scores indicate greater similarity.

The scoring criteria are defined as follows:

- 0 = Completely different reasoning methods and approaches

- 1 = Slightly similar reasoning direction, but different methods

- 2 = Some common reasoning steps, but overall approach is different

- 3 = Similar core approach, but significant differences in specific steps

- 4 = Essentially the same core approach, only minor differences in details or order

- 5 = Essentially the same approach, only different wording

**Importantly, the judge is explicitly instructed to evaluate only the reasoning process and ignore whether the final answers are the same.** This ensures that the diversity score reflects genuine differences in reasoning paths rather than being confounded by answer correctness.

## D.2. Evaluation Prompt

We use the following prompt template for the LLM judge:

*Listing 3.* Diversity Evaluation Prompt

```
Please analyze and compare the similarity of the following two reasoning
    processes.

Reasoning process 1:
{text1}

Reasoning process 2:
{text2}

Please follow these steps for analysis:

Step 1: Extract the core steps of reasoning process 1
Please summarize and extract the main reasoning steps and final answer of
    reasoning process 1, listing the key points concisely.

Step 2: Extract the core steps of reasoning process 2
Please summarize and extract the main reasoning steps and final answer of
    reasoning process 2, listing the key points concisely.

Step 3: Comparative analysis
Compare the two reasoning processes on:
1. Whether the core approaches are the same
2. The degree of similarity in reasoning steps

Step 4: Provide similarity score
Based on the following rating criteria, give a similarity score from 0 to 5:
- 0 = Completely different reasoning methods and approaches
- 1 = Slightly similar reasoning direction, but different methods
- 2 = Some common reasoning steps, but overall approach is different
- 3 = Similar core approach, but significant differences in specific steps
- 4 = Essentially the same core approach, only minor differences in details or
    order
- 5 = Essentially the same approach, only different wording

Important Note: When scoring, do not consider whether the answers are the same!

Please output the final score in the last line, in the format: [Final Score: X] (
    X is an integer from 0 to 5)
```

## D.3. Example: Similarity Matrices on an AIME Problem

We illustrate the evaluation process using a challenging problem from the AIME-120 hard set for Qwen3-30B-A3B-Instruct. We generate 32 responses using three different sampling methods—normal-temperature token sampling, high-temperature token sampling, and Expert-Sample—and compute the pairwise similarity matrix for each method.

Figure 9(a) shows the similarity matrix for normal-temperature token sampling. The matrix exhibits predominantly high similarity scores (shown in yellow, indicating scores of 4–5), suggesting that most response pairs follow nearly identical reasoning paths. This reflects the limited diversity introduced by normal-temperature sampling.

Figure 9(b) shows the similarity matrix for high-temperature token sampling. Compared to normal-temperature sampling,

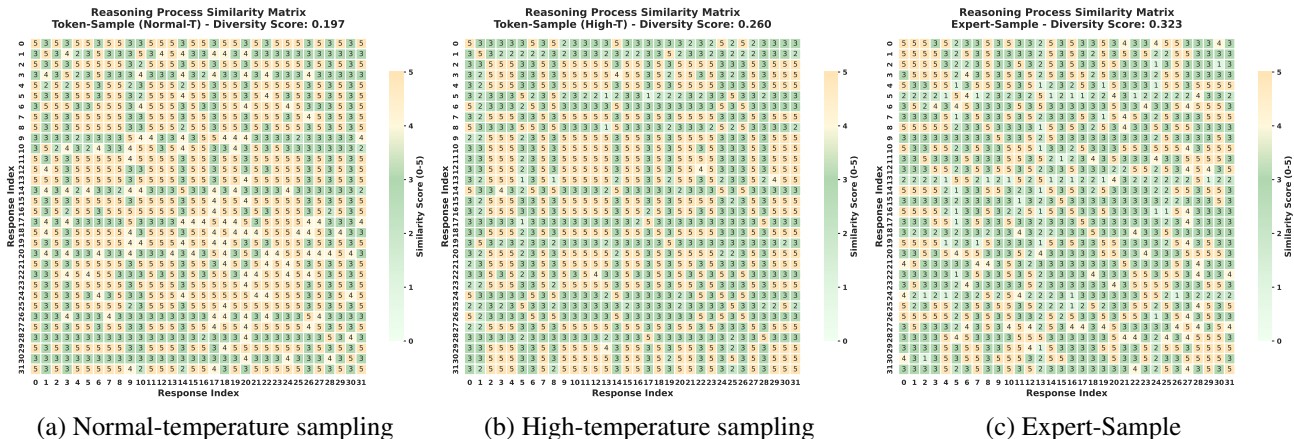

| (a) Normal-temperature sampling | (b) High-temperature sampling | (c) Expert-Sample |

*Figure 9.* Pairwise reasoning path similarity matrices for 32 responses generated by Qwen3-30B-A3B (Instruct) on an AIME-120 hard problem. Lower scores (purple/blue) indicate more diverse reasoning paths, while higher scores (yellow) indicate similar paths. Expert-Sample exhibits substantially more diversity compared to both token-level sampling methods.

we observe more variation in the similarity scores, with some response pairs showing lower similarity (scores of 2–3). However, large clusters of highly similar responses still persist.

Figure 9(c) shows the similarity matrix for Expert-Sample. The matrix displays substantially more diversity, with a broader distribution of similarity scores and notably more low-similarity pairs (scores of 1–3, shown in purple and blue). This indicates that Expert-Sample successfully induces more diverse reasoning paths across the sampled responses.

### D.4. Computing the Diversity Score

Given the $n \times n$ pairwise similarity matrix $S$ where $S_{ij} \in \{0, 1, 2, 3, 4, 5\}$ represents the similarity score between response $i$ and response $j$, we compute the diversity score as follows:

1. Compute the average similarity across all off-diagonal pairs:

$$\text{avg\_similarity} = \frac{1}{n(n-1)} \sum_{i \neq j} S_{ij} \tag{7}$$

2. Normalize the average similarity to $[0, 1]$:

$$\text{normalized\_similarity} = \frac{\text{avg\_similarity}}{5} \tag{8}$$

3. Define the diversity score as:
$$\text{diversity\_score} = 1 - \text{normalized\_similarity} \tag{9}$$

A higher diversity score indicates greater variation in reasoning paths across the sampled responses.

For the example problem shown above, the diversity scores for the three sampling methods are:

- Normal-temperature token sampling: 0.1968

- High-temperature token sampling: 0.2597

- Expert-Sample: **0.3226**

Expert-Sample achieves the highest diversity score, consistent with the visual patterns observed in the similarity matrices. This demonstrates that Expert-Sample effectively induces more diverse reasoning paths compared to token-level sampling methods.

### D.5. Summary

Through this LLM-based evaluation framework, we effectively assess the diversity of reasoning paths across multiple sampled responses. By instructing the judge to focus solely on reasoning processes while ignoring final answers, we obtain a reliable measure of how differently the model approaches the same problem under different sampling strategies. The results consistently show that Expert-Sample produces substantially more diverse reasoning paths than both normal-temperature and high-temperature token sampling.

## E. Detailed Router Weight Distribution Analysis

In Section 2.2, we presented the average router weight distribution of Qwen3-30B-A3B-Instruct across AIME-120, GPQA-Diamond, and LiveCodeBench-V6-lite, revealing that the top-ranked experts exhibit relatively dispersed weight distributions while the remaining experts—including those not selected—have nearly uniform weights, suggesting that the model is not as confident in its selections as one might expect. Here, we provide additional weight distribution results across different datasets and different models to more comprehensively demonstrate this phenomenon.

We recorded the expert selection weights for four models: Qwen3-30B-A3B-Instruct, Ling-lite-1.5-2507, GPT-OSS-20B, and Qwen3-Next-80B-A3B-Instruct. For each model, we collected weights across three diverse tasks: AIME-120 (mathematical reasoning), GPQA-Diamond (professional knowledge reasoning), and LiveCodeBench-V6-lite (code generation). The weights are aggregated across all samples in each dataset, all tokens from both the prefill and decode phases, and all MoE layers in the model. We then generated a composite figure with a $4 \times 2$ grid of subplots for each model. The first three rows correspond to results on the three individual datasets, while the fourth row shows the averaged results across all datasets. In each row, the left subplot displays the weight distribution of all experts ranked by their weights, and the right subplot zooms in on the top-$4k$ experts (where $k$ is the default number of selected experts) to provide a more detailed view of the certain-head and uncertain-tail distribution pattern.

As shown in Figures 10–13, several consistent patterns emerge. First, for any given model, although minor variations exist across different datasets, the overall distribution characteristics remain consistent: a sharp transition separates the high-weight "certain selection" experts from the low-weight "uncertain selection" experts, with the latter forming a nearly flat plateau. Second, this pattern generalizes across models with vastly different architectures: Qwen3-30B-A3B-Instruct with 128 experts selecting 8, Ling-lite-1.5 with 64 experts selecting 6, GPT-OSS-20B with 32 experts selecting 4, and Qwen3-Next-80B-A3B-Instruct with 512 experts selecting 10. Despite these differences in the total number of experts and the selection ratio, all models exhibit the same characteristic: the router confidently selects only a small subset of experts (typically 2–4), while the remaining selected experts have weights comparable to those of unselected candidates. This observation reinforces our motivation that the boundary between selected and unselected experts is often arbitrary, and introducing controlled stochasticity through Expert-Sample can effectively explore this uncertain region to enhance output diversity.

## F. Evaluation Details

### F.1. Evaluation Framework

We use LightEval (version 0.9.1) as our evaluation framework with vLLM (version 0.10.2) as the inference backend. All experiments are conducted on 8 NVIDIA A800-80G GPUs.

### F.2. Model Details

All models used in our experiments are the latest available versions at the time of writing. Specifically, Qwen3-30B-A3B-Instruct and Ling-Lite-1.5 refer to their July 2025 (2507) checkpoint releases. For GPT-OSS-20B, we set the reasoning effort to "low" in all experiments.

### F.3. Dataset Details

We evaluate on seven benchmarks across our experiments: AIME-120, AIME-2024, AIME-2025, HMMT-2025, MATH-500-Hard, GPQA-Diamond, and LiveCodeBench-V6-Lite.

For AIME-2024, AIME-2025, and HMMT-2025, each dataset contains only 30 samples. To ensure statistical reliability, we

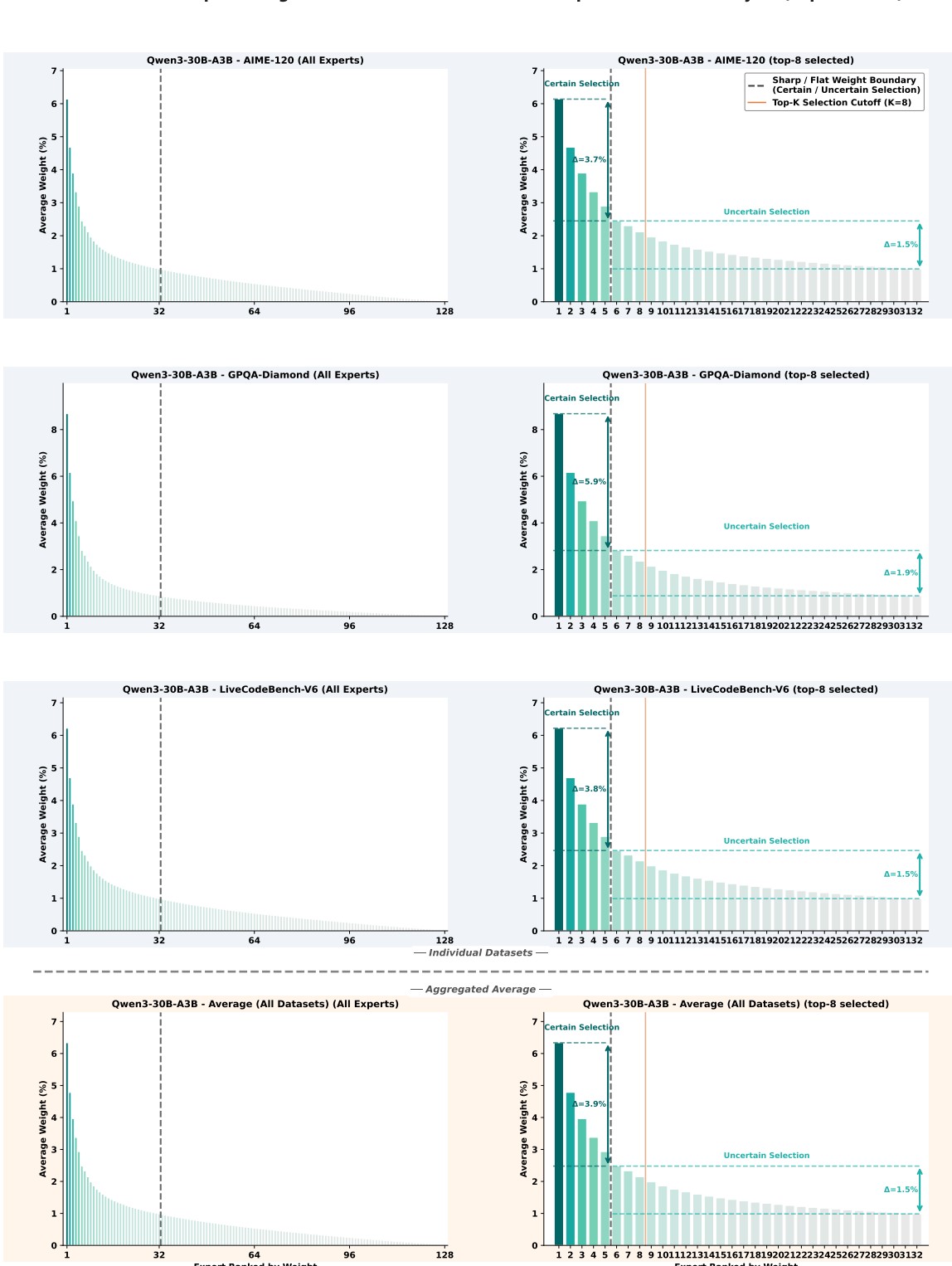

*Figure 10.* Router weight distribution analysis for Qwen3-30B-A3B (Instruct) across different datasets. Left column: full expert weight distribution ranked by weight. Right column: detailed view of top-$4k$ experts showing the certain selection (high-weight) and uncertain selection (low-weight) regions.

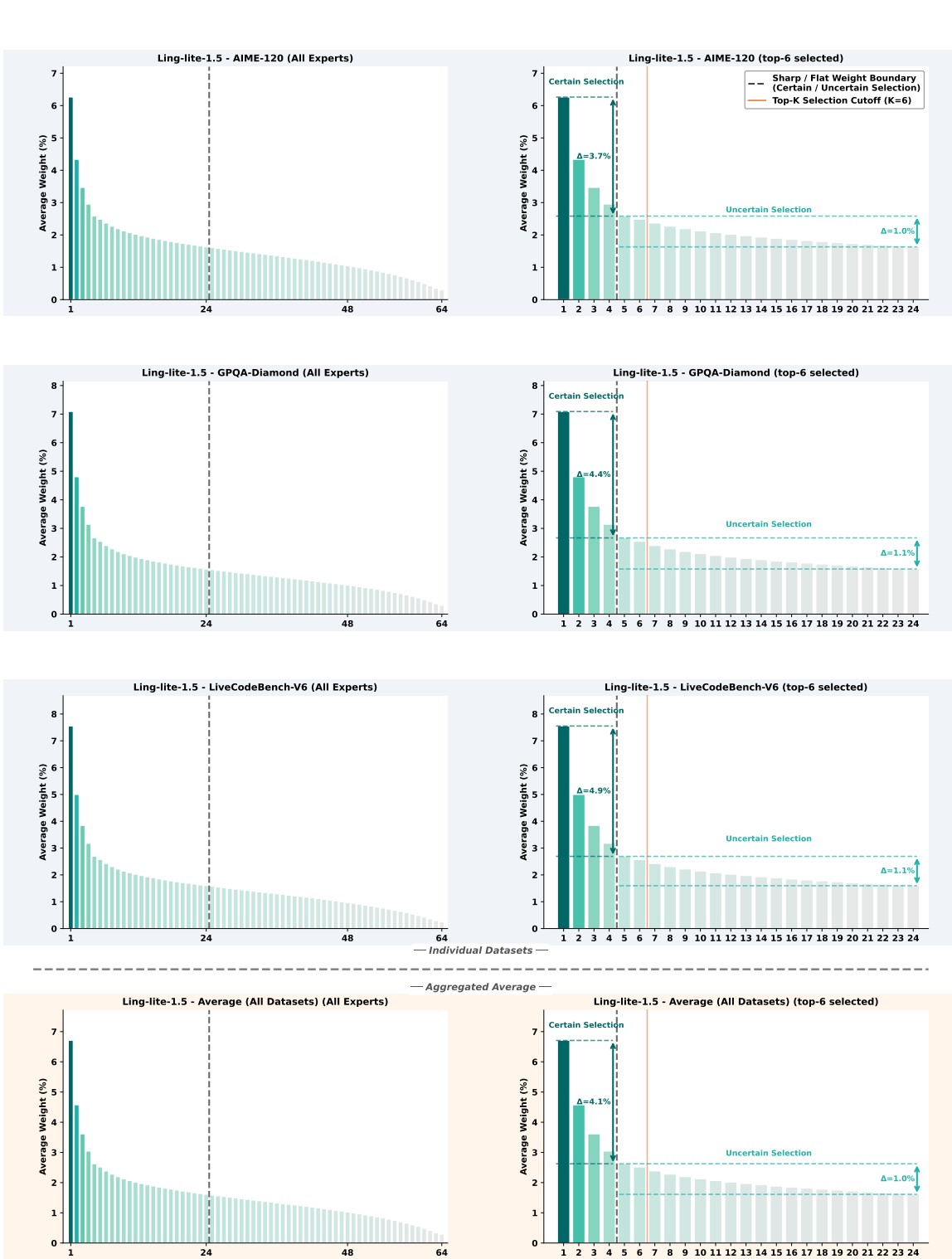

*Figure 11.* Router weight distribution analysis for Ling-lite-1.5 across different datasets.

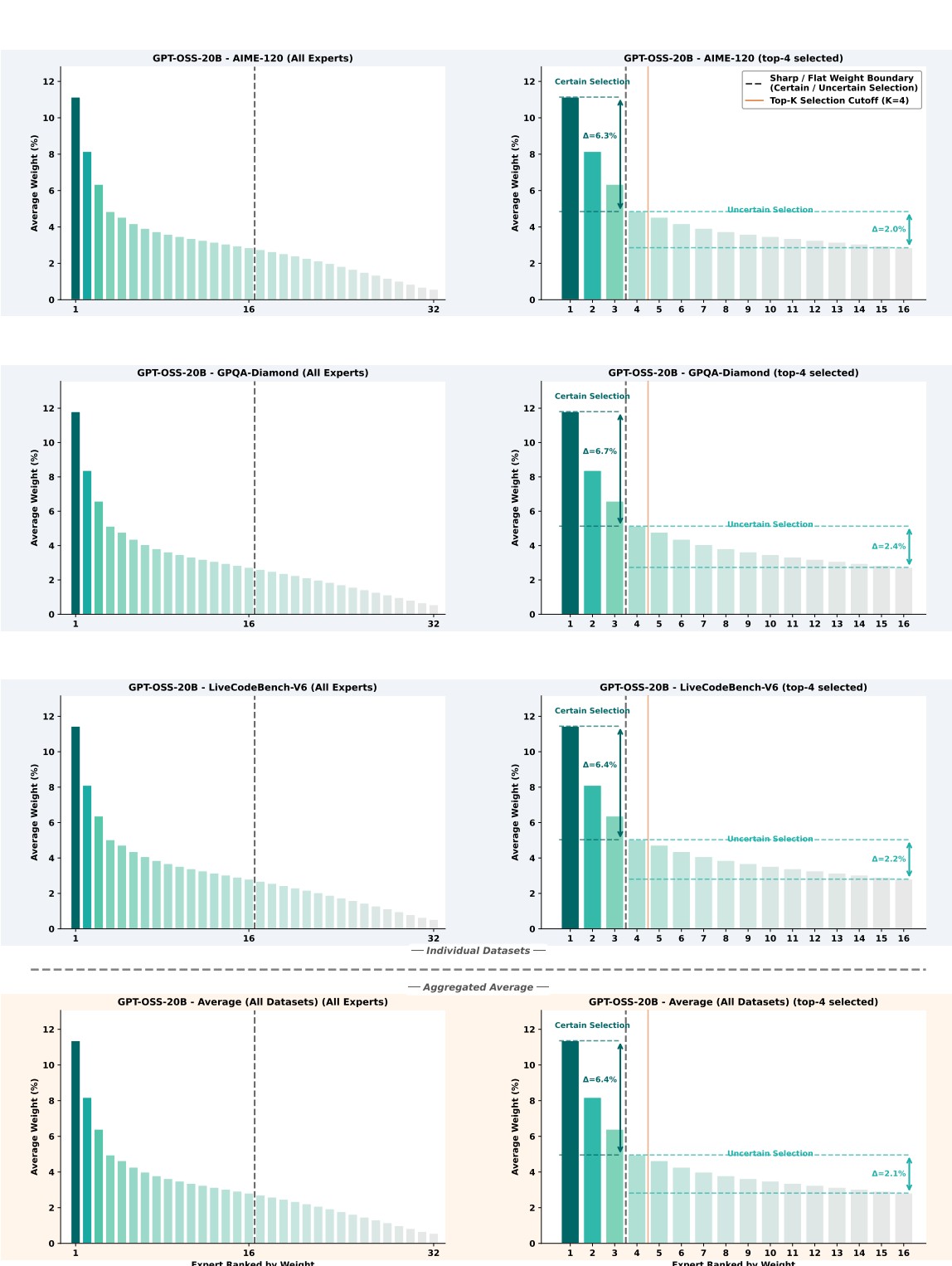

*Figure 12.* Router weight distribution analysis for GPT-OSS-20B across different datasets.

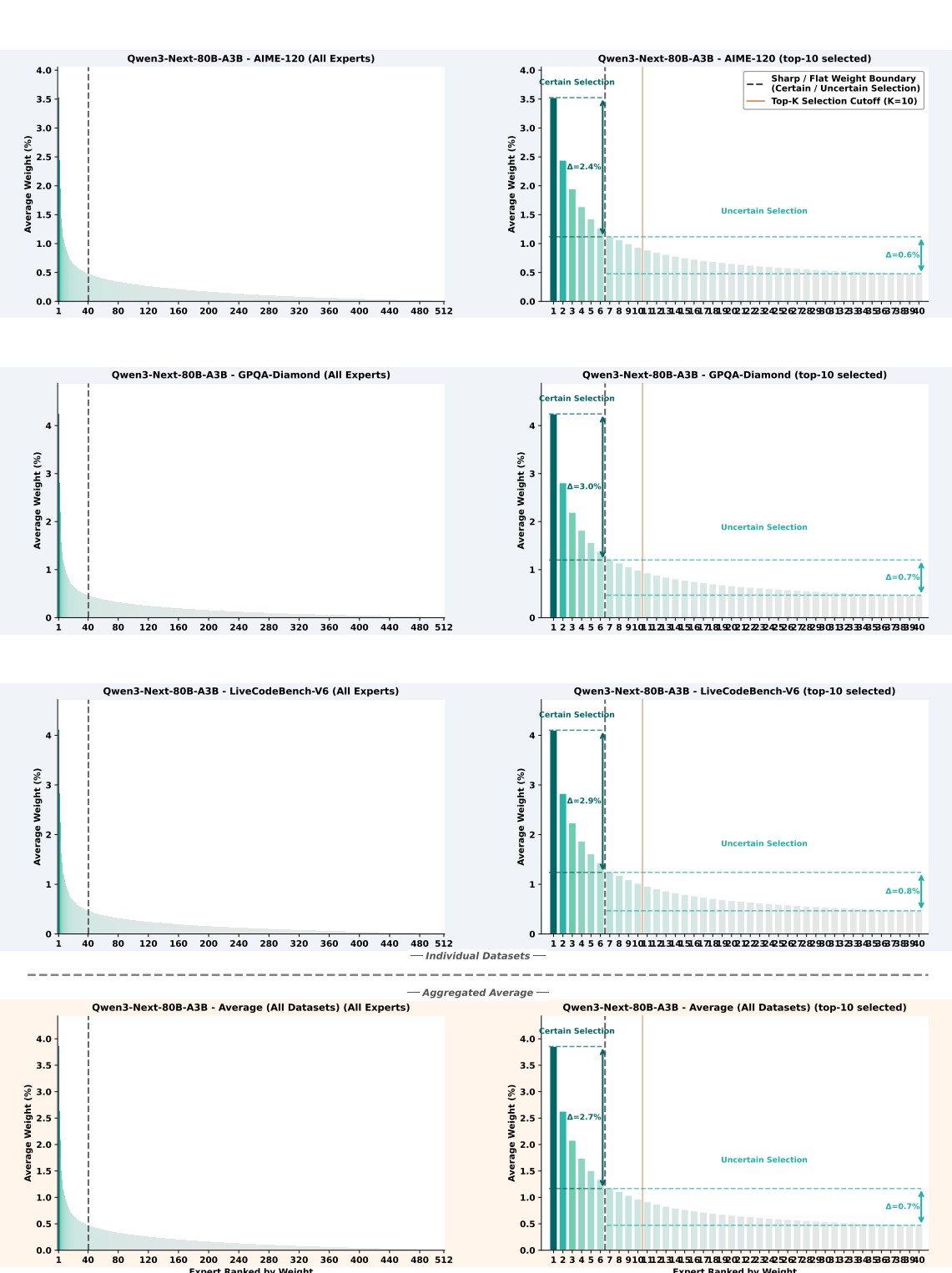

*Figure 13.* Router weight distribution analysis for Qwen3-Next-80B-A3B (Instruct) across different datasets.

run 5 independent trials and report the average accuracy for these benchmarks.

For HMMT-2025, the ground-truth answers have complex formats that often cause conventional answer-matching tools to produce false negatives. To address this, we employ an LLM-as-judge approach: specifically, we use Qwen2.5-7B-Instruct as the judge model to determine the correctness of each response by comparing the model's answer against the ground truth.

### F.4. Prompts

We provide the prompt templates used for each dataset below.

### AIME-2024 / AIME-2025 / AIME-120 / HMMT-2025 / MATH-500-Hard

```
{question}
Remember to put your final answer within \boxed{}.
```

### GPQA-Diamond

```
Answer the following multiple choice question. The last line of your response
    should be of the following format: 'ANSWER: $LETTER' (without quotes) where
    LETTER is one of ABCD. Think step by step before answering.
{question}
A) {A}
B) {B}
C) {C}
D) {D}
```

### LiveCodeBench-V6-Lite

```
You are an expert Python programmer. You will be given a question (problem
    specification) and will generate a correct Python program that matches the
    specification and passes all tests. You will NOT return anything except for
    the program.
{question}
```

## G. Related Work

### G.1. Test-Time Scaling

Test-time scaling improves model performance by generating multiple candidates to cover the correct solution and selecting the correct one among them. One line of research focuses on enhancing diversity in multi-sample generation, either through dynamic temperature adjustment (Zhang et al., 2024) or by rephrasing questions (Zhou et al., 2024) and modifying prompts (Naik et al., 2024). Another line of work focuses on guiding search and verifying results, including beam search (Guo et al., 2024), tree-based reasoning (Yao et al., 2023), and learned or rule-based verifiers (Irvine et al., 2023). Overall, the effectiveness of test-time scaling depends on two factors: whether the correct answer is covered among multiple samples, and whether it can be successfully identified. We focus on the former aspect, aiming to enhance reasoning diversity to improve coverage of correct solutions, which also serves as the foundation for the latter.

### G.2. Fine-Grained MoE

With a substantially larger pool of well-trained experts, fine-grained MoE offers greater flexibility in expert selection. Numerous works (Lu et al., 2024; Huang et al., 2025) explore expert pruning to improve inference efficiency. Others manually adjust expert selection to enhance specific capabilities, such as strengthening critical experts to boost reasoning (Wang et al., 2025). These works collectively suggest greedy top-$k$ selection learned during pretraining may not be optimal, leaving significant potential for better utilizing the large expert pool at inference time.

