# OpenReview forum: "Certain Head, Uncertain Tail: Expert-Sample for Test-Time Scaling in Fine-Grained MoE"
_ICML.cc/2026/Conference — ICML 2026 regular_

### Official Review · Reviewer_vz6W · 2026-03-13

**Soundness:** 3
**Presentation:** 3
**Significance:** 2
**Originality:** 2
**Overall Recommendation:** 5
**Confidence:** 3

**Summary:**

Diversity and stability are typically traded off by increasing/decreasing the temperature. In practice, the MoE router selects the top k experts instead of sampling across the expert distribution. This offers a new opportunity modulate diversity and stability by controlling which experts are selected. Since multiple experts are selected, this offers much more fine-grained control: pick some of the top experts to promote stability and some of the tail experts to promote diversity. This offers better control over the tradeoff, improving performance at larger k for pass@k.

**Compliance With Llm Reviewing Policy:**

Affirmed.

**Key Questions For Authors:**

1. It could be nice to have a more detailed temperatue tradeoff (i.e. show pass@k for many temperatures overlayed) as a baseline for Expert-Sample to show that it dominates all temperatures. Another way to visualize this is x-axis diversity and y-axis average performance for a fixed number of generations. This is mostly a visual suggestion: it seems that Expert-Sample would beat it.

**Limitations:**

Yes

**Strengths And Weaknesses:**

Strengths
1. The proposed method is a novel interpretation of the MoE router as a decoding algorithm
2. The method seems to improve pass@k performance
3. The work evaluates the method across a wide range of models and benchmarks, giving confidence to the performance improvements

Weaknesses
1. The scope and performance improvements are limited, which is not a weakness but a reason for not giving a higher score.

---

> ### Author Rebuttal · Authors · 2026-03-30
>
> Dear reviewer vz6W,
>
> We sincerely thank the reviewer for the time and effort spent on our paper. We are encouraged by the recognition of our work.
>
> **Regarding the weakness:** We appreciate the reviewer's understanding. Since Expert-Sample is plug-and-play and training-free, it yields notable improvements in scaling (pass@k) and relatively limited gains in end-to-end accuracy when combined with verification. We believe this is because our method does not specifically guide the search process — its benefit to verification comes from improved diversity of sampling paths. In future work, we plan to explore more fine-grained and controlled expert-sample strategies to pursue more significant improvements across the full test-time scaling pipeline.
>
> **Regarding the key question:** We thank the reviewer for this excellent visualization suggestion. Following this suggestion, we plot the pass@k scaling curves of Token-Sample under various temperatures ($t \in \{0.3, 0.5, 0.7, 0.9, 1.1, 1.3\}$, with top-p=0.95 and top-k=20) alongside Expert-Sample, using Qwen3-30B-A3B-Instruct on GPQA-Diamond as an example. The figure is presented as Figure R1 at [[anonymous link]](https://anonymous.4open.science/r/Rebuttal-ES-CAF7). As shown in the figure, this visualization clearly illustrates that Expert-Sample surpasses Token-Sample across all sampling temperatures throughout the entire scaling range.

---

> > ### Author Rebuttal · Reviewer_vz6W · 2026-04-03
> >
> > My questions are in a separate comment.

---

> > > ### Author Response · Authors · 2026-04-07
> > >
> > > We sincerely thank you for your time and positive assessment of our work. We greatly value your feedback, and your insights have been very helpful in improving our paper. In your rebuttal acknowledgement, you mentioned "My questions are in a separate comment." **However, as the discussion period approaches its end, we have been unable to find any visible comment or follow-up questions from you.** We are concerned that you may have posted an official comment that was not set to be visible to authors.
> > >
> > > We take your concerns very seriously and would not want to miss the opportunity to address them. If you have any follow-up questions, we would be happy to discuss them with you in the remaining time. Since we only have one rebuttal submission opportunity, we will address your questions by editing this response.

---

### Official Review · Reviewer_sLwz · 2026-03-13

**Soundness:** 4
**Presentation:** 3
**Significance:** 2
**Originality:** 2
**Overall Recommendation:** 4
**Confidence:** 3

**Summary:**

They empirically characterize fine-grained MoE routing and uncover an informative pattern: router scores exhibit a certain head of high-confidence experts followed by an uncertain tail of low-confidence candidates. Motivated by these findings, they propose Expert-Sample, a training-free method that preserves high-confidence selections while injecting controlled stochasticity into the uncertain tail, enabling diverse generation without destabilizing outputs. Evaluated on multiple fine-grained MoE models across math, knowledge reasoning, and code tasks, Expert-Sample consistently improves pass@n and verification-based accuracy.

**Compliance With Llm Reviewing Policy:**

Affirmed.

**Final Justification:**

My main concerns have been addressed. I have raised my score.

For W1, I encourage the authors to clarify that the 1% overhead is not regarding the total cost but the "per-path cost."

**Key Questions For Authors:**

See weaknesses.

**Limitations:**

See weaknesses.

**Strengths And Weaknesses:**

Strengths:

S1 (Performance Gains): Expert-Sample outperforms baseline methods in pass@n and verification-based accuracy.

S2 (Training-Free Integration): Expert-Sample acts as a plug-and-play sampling strategy that requires no additional training.

S3 (Insightful Observation): Router scores exhibit a certain head of high-confidence experts followed by an uncertain tail of low-confidence candidates.

Weaknesses:

W1 (Inconsistent Claims): While this paper claims Expert-Sample as a test-time scaling method, Section 5 claims that the computational overhead remains within 1%. This seems to suggest that either this method is not effectively scaling test-time compute or the overhead analysis is misleading.

W2 (Limited Novelty): I searched on Google and immediately found a similar method [1] that also scales test-time compute through experts in MoE. However, this paper has no discussion on such similar work.

[1] https://arxiv.org/pdf/2509.22572

W3 (No Greedy Baseline): The paper did not provide greedy decoding as a baseline. Hence, it is unclear how scaling-efficient the proposed method Expert-Sample is compared with no scaling (greedy).

W4 (Weak Baselines): The compared baseline methods seem rather weak (only one recent method plus two standard approaches). It is strongly encouraged to compare Expert-Sample with more recent baseline methods in test-time scaling.

W5 (Many Hyperparameters): Expert-Sample introduces three hyperparameters (Section 3.2). While the paper gives a recommended setting (Section 3.4), it still does not seem easy to tune these hyperparameters in practice.

---

> ### Author Rebuttal · Authors · 2026-03-30
>
> Dear reviewer sLwz,
>
> We sincerely thank the reviewer for the constructive and detailed feedback. We address each concern below.
>
> > **W1.** (Inconsistent Claims) ...
>
> Thanks for raising this point. We would like to clarify: the "within 1%" overhead in Section 5 refers to the **per-path cost** introduced by our stochastic routing mechanism. That is, for a single forward pass, the additional computation from our modified router is negligible.
>
> This does not suggest that Expert-Sample avoids the cost of sampling multiple paths. The key advantage of Expert-Sample is that it achieves greater diversity across these paths without meaningful per-path overhead. We will revise the manuscript to make this distinction clearer.
>
> ---
>
> > **W2.** (Limited Novelty) ...
>
> We thank the reviewer for bringing this concurrent work to our attention. We will cite and discuss DES[1] in the revised version. While the two methods share some high-level motivation, they differ fundamentally in perspective and design.
>
> Expert-Sample introduces a **new sampling dimension** for TTS: expert-level sampling operates on the MoE router, orthogonal to and composable with standard token-level sampling on the LM head. By injecting structured stochasticity into expert selection at every token, it enriches the sampling space and raises the upper bound of scaling while keeping the number of activated experts unchanged. DES, in contrast, varies the number of activated experts (the value of k in top-k routing) across reasoning paths as a macro-level hyperparameter, and is tightly coupled with beam search and a process reward model (PRM). The two methods thus operate at different stages: DES integrates MoE structure into search and selection, while Expert-Sample targets sample generation.
>
> This leads to a practical difference: Expert-Sample is agnostic to the search strategy and serves as a **plug-and-play, training-free** complement to any sampling-based TTS pipeline, whereas DES requires specific search infrastructure. We will add a direct empirical comparison with DES in our response to W4.
>
> ---
>
> > **W3.** (No Greedy Baseline) & **W4.** (Weak Baselines) ...
>
> As discussed above, Expert-Sample provides a complementary sampling dimension (expert-level) rather than a new token-level strategy. Since it does not conflict with either token-sampling methods or search/selection methods, **it is orthogonal to most existing TTS approaches — this is the primary reason we did not compare against many other methods.** Nevertheless, we agree that more baselines would be beneficial. Following the reviewer's suggestion, we supplement experiments from both stages: sample generation (diversity-enhancing methods) and sample selection (verification/search methods), with greedy decoding included as the no-scaling baseline (addressing W3).
>
> **(1) Scaling Generation.** We compare against AdapT[2] and SCoP[3], two methods that also enhance sampling diversity at the generation stage. Experiments are conducted with Qwen3-30B-A3B-Instruct on AIME-120 (greedy accuracy: 68.33).
>
> | Method | pass@2 | pass@4 | pass@8 | pass@16 | pass@32 | pass@64 |
> |---|---|---|---|---|---|---|
> | AdapT | 76.67 | 79.17 | 83.33 | 86.67 | 87.5 | 89.17 |
> | SCoP | 75.83 | 80 | 84.17 | 85.83 | 87.5 | 88.33 |
> | **Expert-Sample (Ours)** | **79.17** | **82.5** | **87.5** | **88.33** | **90** | **91.67** |
>
> **(2) With Verification .** We combine Expert-Sample with Weighted Majority Voting, a standard selection method, and compare against DES[1] and DVTS[4] which employ guided search with PRMs. Experiments are conducted with Qwen3-30B-A3B-Instruct and results are reported at N=32.
>
> | Method | HumanEval | AIME 2024 | AIME 2025 |
> |---|---|---|---|
> | Greedy | 88.41 | 70.00 | 66.67 |
> | DES | 94.51 | 86.67 | 70.00 |
> | DVTS | 93.90 | 86.67 | 70.00 |
> | **Expert-Sample (Ours)** | **95.73** | **91.33** | **76.67** |
>
> ---
>
> > **W5.** (Many Hyperparameters) ...
>
> We appreciate this concern. In Appendix C, we provide a detailed hyperparameter analysis across three subsections, each dedicated to one of the three hyperparameters ($k_{\text{keep}}$, temperature $t$, and sampling range). The results on both Qwen3-30B-A3B-Instruct and Ling-lite-1.5 consistently show that Expert-Sample is robust to a wide range of hyperparameter choices — performance remains stable without careful tuning. The recommended setting in Section 3.4 was used across all benchmarks and models in our paper without any task-specific adjustment. We will move key findings from Appendix C to the main text in the revision to make this robustness more prominent.
>
>
> **References:**
>
> [1] Dynamic Experts Search: Enhancing Reasoning in Mixture-of-Experts LLMs at Test Time.
>
> [2] Hot or Cold? Adaptive Temperature Sampling for Code Generation with Large Language Models.
>
> [3] Paraphrase and Solve: Exploring and Exploiting the Impact of Surface Form on Mathematical Reasoning in Large Language Models.
>
> [4] ETS: Efficient Tree Search for Inference-Time Scaling.

---

> > ### Author Rebuttal · Reviewer_sLwz · 2026-04-03
> >
> > I thank the authors for the detailed rebuttal. I have raised my score.
> >
> > For W1, I encourage the authors to clarify that the 1% overhead is not regarding the total cost but the "per-path cost."

---

> > > ### Author Response · Authors · 2026-04-04
> > >
> > > We sincerely thank the reviewer for raising the score and for the helpful suggestion. We will clearly clarify in the revised paper that the 1% overhead refers to the per-path cost rather than the total cost, to avoid any potential misunderstanding.

---

### Official Review · Reviewer_m237 · 2026-03-13

**Soundness:** 2
**Presentation:** 3
**Significance:** 3
**Originality:** 3
**Overall Recommendation:** 4
**Confidence:** 3

**Summary:**

The paper proposes Expert-Sample, a training-free inference strategy for improving test-time scaling in fine-grained Mixture-of-Experts (MoE) models. Motivated by the empirical finding that router scores exhibit a high-confidence “certain head” and a relatively flat “uncertain tail,” the method deterministically preserves top-ranked experts to maintain stability while stochastically sampling the remaining experts to encourage reasoning diversity. Experiments on math, coding, and knowledge reasoning benchmarks show that this routing-level intervention improves pass@n and verification-based accuracy across several fine-grained MoE models, while adding negligible runtime overhead.

**Compliance With Llm Reviewing Policy:**

Affirmed.

**Final Justification:**

My main concerns have been addressed. I raise my score to 4 and encourage the authors to incorporate the new experiments into the revised paper.

**Key Questions For Authors:**

See weaknesses

**Limitations:**

yes

**Strengths And Weaknesses:**

## Strengths:
1. The empirical observation is solid and well-supported. The asymmetry between greedy accuracy and pass@n under expert reduction is demonstrated across five different models, providing convincing motivation for the method.

2. The method is simple and the evaluation is comprehensive, both pass@n scaling and verification experiments. The three-tier difficulty split (Correct/Medium/Hard) is a nice design for separately validating stability and diversity.

3. Overhead analysis is thorough, and hyperparameter ablations confirm robustness of the default configuration, supporting the plug-and-play claim.

## Weaknesses:

1. The mechanism behind the “uncertain tail” remains under-justified. The paper argues that flat-scored tail experts contribute useful diversity, but does not disentangle whether these experts are genuinely complementary or simply weakly relevant. A stronger analysis would compare Expert-Sample against simpler routing-side perturbation baselines, such as uniform tail sampling or noisy routing without an explicit head/tail split, to verify that the gains come from structured routing exploration rather than generic stochasticity.

2. The evaluation is concentrated on exploration-friendly tasks. All experiments are conducted on math, knowledge reasoning, or code benchmarks, where repeated sampling is naturally beneficial. Given the paper’s drop-in / general-purpose framing, it would be important to test whether routing perturbations preserve quality on tasks such as instruction following or open-ended generation, where diversity is not the only concern.

3. The diversity evaluation is under-validated. Appendix D relies on a single LLM judge (DeepSeek-R1) for pairwise reasoning similarity, but does not report judge consistency, human validation, or robustness to alternative judges. Since diversity is central to the paper’s narrative, more objective or triangulated analyses would strengthen this aspect of the analysis.

---

> ### Author Rebuttal · Authors · 2026-03-30
>
> Dear reviewer m237,
>
> We sincerely thank the reviewer for the constructive and detailed feedback. We address each concern below.
>
> > **W1.** The mechanism behind the "uncertain tail" remains under-justified.
>
> We agree that disentangling structured routing exploration from generic stochasticity is valuable. We address this through both existing evidence in our paper and newly supplemented experiments.
>
> **Existing evidence.**
>
> - **$k_{\text{keep}}$ ablation** (Appendix C, Figures 5–6): Setting $k_{\text{keep}}=0$ is exactly the "no head/tail split" scenario raised by the reviewer. Under this setting, Correct Set accuracy drops sharply (91.5%→72.8%) while Uncertain Set gains diminish, showing that unstructured stochasticity hurts stability without yielding effective diversity.
>
> **New baselines.** Following the reviewer's suggestion, we further compare two routing-side perturbation baselines on full datasets — Qwen3-30B-A3B-Instruct across AIME120, GPQA-Diamond, and LiveCodeBench-v6:
> - **Uniform-Tail:** head fixed (top-5), tail sampled uniformly ignoring routing scores.
> - **Noisy-Full-Routing:** Gumbel noise added to all expert logits, then standard top-k without head/tail split.
>
> | Method | AIME120 pass@32 | AIME120 pass@64 | GPQA pass@32 | GPQA pass@64 | LCB-v6 pass@32 | LCB-v6 pass@64 |
> |---|---|---|---|---|---|---|
> | Token-Sample (Baseline) | 87.5 | 87.5 | 85.35 | 89.9 | 60 | 63.43 |
> | **Expert-Sample (Ours)** | **90** | **91.67** | **91.92** | **93.43** | **64.57** | **68.57** |
> | Uniform-Tail | 85 | 85.83 | 83.84 | 89.39 | 57.14 | 59.43 |
> | Noisy-Full-Routing | 78.33 | 80 | 77.78 | 83.84 | 56.57 | 57.14 |
>
> Both consistently underperform even the standard Token-Sample baseline, confirming that naïve randomness in routing is harmful. Only Expert-Sample's structured head/tail design yields meaningful gains, corroborating the existing evidence above. Together, these results confirm that the gains of Expert-Sample indeed come from structured routing exploration rather than generic stochasticity.
>
> ---
>
> > **W2.** The evaluation is concentrated on exploration-friendly tasks.
>
> We appreciate this suggestion. Our original benchmark selection follows the established practice in test-time scaling research, which predominantly focuses on tasks with verifiable answers where pass@k is naturally well-defined (open-ended generation lacks clear correctness criteria, making it difficult to properly evaluate and present scaling results). Nevertheless, we agree that evaluating on instruction-following tasks would strengthen our general-purpose claim. We therefore supplement experiments on IFEval and IFBench with Qwen3-30B-A3B-Instruct:
>
> | Method | Benchmark | pass@2 | pass@4 | pass@8 | pass@16 | pass@32 | pass@64 |
> |---|---|---|---|---|---|---|---|
> | Token-Sample | IFEval | 86.51 | 88.72 | 90.02 | 90.57 | 91.68 | 92.24 |
> | **Expert-Sample (Ours)** | IFEval | **89.09** | **90.57** | **91.87** | **91.87** | **93.90** | **94.82** |
> | Token-Sample | IFBench | 35.33 | 38.67 | 39.67 | 42.00 | 43.00 | 44.33 |
> | **Expert-Sample (Ours)** | IFBench | **38.33** | **41.00** | **42.00** | **43.33** | **45.67** | **47.00** |
>
> Expert-Sample consistently outperforms standard token-level sampling and achieves more efficient scaling, confirming it is an effective drop-in enhancement beyond reasoning-centric benchmarks.
>
> ---
> > **W3.** The diversity evaluation is under-validated.
>
> We appreciate this suggestion and agree that more comprehensive diversity evaluation would strengthen this aspect. Due to time constraints, we focus on reporting judge consistency and robustness to alternative judges.
>
> **(1) Judge consistency.** We re-run the DeepSeek-R1 pairwise reasoning similarity evaluation three times on the Uncertain Set of Qwen3-30B-A3B-Instruct (AIME120) and report all results:
>
> | Method | Run 1 | Run 2 | Run 3 | Avg ± Std |
> |---|---|---|---|---|
> | Token-Sample (t=0.7) | 0.214 | 0.229 | 0.241 | 0.228 ± 0.014 |
> | Token-Sample (t=1.3) | 0.295 | 0.291 | 0.303 | 0.296 ± 0.006 |
> | **Expert-Sample (Ours)** | **0.371** | **0.386** | **0.401** | **0.386 ± 0.015** |
>
> **(2) Robustness to alternative judges.** We replace the judge with GPT-4o and Claude-Sonnet-4.6 respectively and conduct the same evaluation (three times). We report **Avg ± Std** below and show per-run scores in Table R1~2 at https://anonymous.4open.science/r/Rebuttal-ES-CAF7
>
> | Method | GPT-4o | Claude-Sonnet-4.6 |
> |---|---|---|
> | Token-Sample (t=0.7) | 0.294 ± 0.007 | 0.258 ± 0.010 |
> | Token-Sample (t=1.3) | 0.349 ± 0.008 | 0.321 ± 0.008 |
> | **Expert-Sample (Ours)** | **0.422 ± 0.005** | **0.403 ± 0.012** |
>
> For judge consistency, although minor fluctuations exist across runs, the standard deviations remain small and the relative gaps between methods are stable. For cross-judge robustness, GPT-4o and Claude-Sonnet-4.6 tend to assign relatively higher diversity scores overall, but relative advantage of Expert-Sample over Token-Sample remains clearly evident, further validating our findings.

---

> > ### Author Rebuttal · Reviewer_m237 · 2026-04-02
> >
> > I thank the authors for the detailed rebuttal. My main concerns have been addressed. I raise my score to 4 and encourage the authors to incorporate the new experiments into the revised paper.

---

> > > ### Author Response · Authors · 2026-04-02
> > >
> > > We sincerely thank the reviewer for the positive acknowledgement and for raising the score. We will incorporate all newly supplemented experiments and analyses into the revised paper.

---

### Official Review · Reviewer_mdDb · 2026-04-03

**Soundness:** 3
**Presentation:** 3
**Significance:** 3
**Originality:** 3
**Overall Recommendation:** 5
**Confidence:** 1

**Summary:**

The paper proposes a training-free method to enable diverse generation without causing the destabilization of outputs that improves performance measured by pass@n and verification-based accuracy across diverse tasks such as code, math, and knowledge reasoning.

**Compliance With Llm Reviewing Policy:**

Affirmed.

**Final Justification:**

To my knowledge, the paper is satisfactory for acceptance based on soundness, originality, significance, and clarity.

**Key Questions For Authors:**

The following are minor clarifications:
1. How sensitive are the reported gains to the choice of verifier?
2. Under what conditions should practitioners prefer Expert-Sample?

**Limitations:**

Yes

**Strengths And Weaknesses:**

1. A key strength is that the paper is based on intuitive, well-motivated empirical observations, which strengthens the soundness and presentation of the paper.
2. Another key strength is that it improves pass@n, best of n, and weighted majority voting while being training-free, which is especially significant.
3. The presentation is clear to my knowledge.

---

### Decision · Program_Chairs · 2026-04-30

**Decision:**

Accept (regular)

**Comment:**

The paper proposes Expert-Sample, a training-free inference strategy for improving test-time scaling in fine-grained Mixture-of-Experts (MoE) models. All reviewers are in agreement to accept the paper and agree with the merits. Please answer the reviewers' questions in the revision. Overall, I recommend acceptance.